# NAIPv2: Debiased Pairwise Learning for Efficient Paper Quality Estimation

**Penghai Zhao**[2,1,3,†], **Jinyu Tian**[2,1,3,†], **Qinghua Xing**[2,1,3,†], **Xin Zhang**[2,1,3], **Zheng Li**[2,1,3]
**Jianjun Qian**[4], **Ming-Ming Cheng**[1,2,3], **Xiang Li**[1,2,3,*]

[1]NKIARI, Shenzhen Futian
[2]VCIP, CS, Nankai University
[3]AAIS, Nankai University
[4]PCALab, School of Computer Science and Engineering,
Nanjing University of Science and Technology

[2]{zhaopenghai,tianjinyu,xingqinghua,zhasion,zhengli97,
cmm,xiang.li.implus}@nankai.edu.cn
[4]csjqian@njust.edu.cn

[†]Equal contribution    [*]Corresponding author

## Abstract

The ability to estimate the quality of scientific papers is central to how both humans and AI systems will advance scientific knowledge in the future. However, existing LLM-based estimation methods suffer from high inference cost, whereas the faster direct score regression approach is limited by translational inconsistencies. We present NAIPv2, a debiased and efficient framework for paper quality estimation. NAIPv2 employs pairwise learning within domain-year groups to reduce inconsistencies in reviewer ratings and introduces the Review Tendency Signal (RTS) as a probabilistic integration of reviewer scores and confidences. To support training and evaluation, we further construct NAIDv2, a large-scale dataset of 24,276 ICLR submissions enriched with metadata and detailed structured content. Trained on pairwise comparisons but enabling efficient pointwise prediction at deployment, NAIPv2 achieves state-of-the-art performance (78.2% AUC, 0.432 Spearman), while maintaining scalable, linear-time efficiency at inference. Notably, on unseen NeurIPS submissions, it further demonstrates strong generalization, with predicted scores increasing consistently across decision categories from Rejected to Oral. These findings establish NAIPv2 as a debiased and scalable framework for automated paper quality estimation, marking a step toward future scientific intelligence systems. Project page is available at
https://sway.cloud.microsoft/Pr42npP80MfPhvj8.

## 1 Introduction

The rapid progress of large language models (LLMs) has enabled a growing ecosystem of AI-for-Science systems, such as autonomous research agents (Lu et al., 2024; Liu et al., 2025; Baek et al., 2024), automated survey-generation pipelines (Wang et al., 2024b; Yan et al., 2025; Liang et al., 2025), and emerging literature-intelligence tools (He et al., 2025; Chen et al., 2025; Li et al., 2025b), etc. A central bottleneck shared by these systems is the need to quickly identify high-quality and early-stage research from the continuous stream of newborn papers. However, such papers lack citation histories, reliable bibliometric signals, and meaningful venue-based indicators. Furthermore, both DoRA (F1000Research et al., 2012) and the Leiden Manifesto (Hicks et al., 2015) caution against using prestige-based metrics as proxies for scientific quality. Consequently, growing attention is directed toward approaches that evaluate research quality based on content-related features rather than external attributes.

Recent work has explored predicting citation-based indicators directly from textual content (Zhao et al., 2025; de Winter, 2024). While these approaches provide useful signals for modeling aspects

of research impact, they are generally oriented toward predicting future citation behavior rather than assessing a paper's intrinsic quality. Prior analyses (Cortes & Lawrence, 2021; Dougherty & Horne, 2022) have noted that citation counts may not reliably track scientific merit, cautioning against their use as a sole indicator of quality. These considerations motivate the development of quality estimation methods that rely more directly on intrinsic, content-grounded signals.

For decades, expert peer review has served as a reliable standard for evaluating research's intrinsic quality. While rigorous, this manual process is costly, time-consuming, and inherently difficult to scale. In response, recent studies (Yu et al., 2024; Zhu et al., 2025; D'Arcy et al., 2024; Idahl & Ahmadi, 2024) have exploited LLMs to assist in the review process (e.g., generating review comments, predicting review scores, or inferring acceptance decisions directly from paper content). As shown in Fig. 1 left panel, although these approaches yield more interpretable textual explanations, their reliance on autoregressive reasoning over long contexts incurs substantial latency and computational overhead. In certain settings, the absence of accessible PDFs renders them entirely inoperative, thereby constraining their applicability in many academic search and recommendation systems.

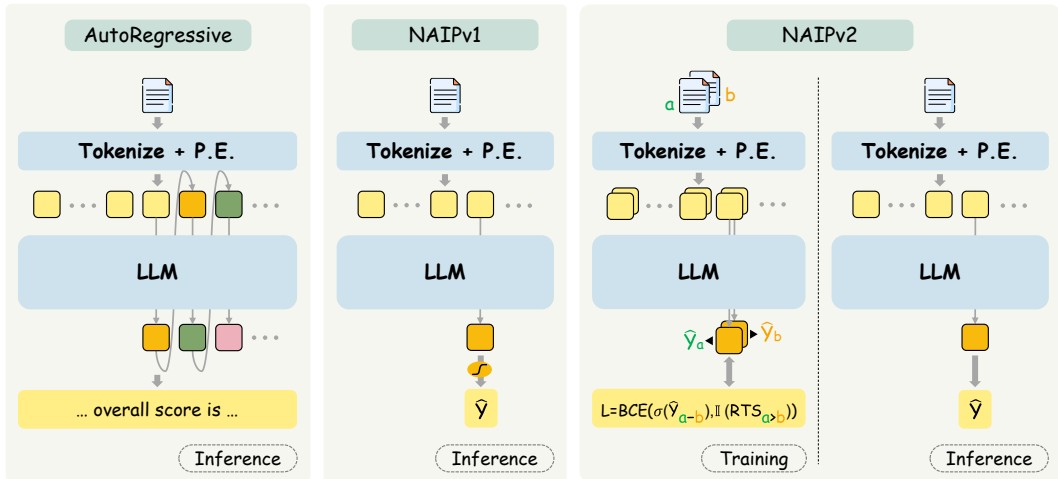

Figure 1: Comparison of various frameworks. AutoRegressive approaches (Zhu et al., 2025; Weng et al., 2025) rely on sequential generation, resulting in substantial inference latency; NAIPv1 (Zhao et al., 2025) enables fast pointwise regression but suffers from translational inconsistency; NAIPv2 leverages debiased pairwise training with confidence-aware signals and operates as an efficient pointwise regressor with linear-time complexity at inference.

To reduce the inference latency inherent in autoregressive approaches, we revisit regression-based methods, i.e., NAIPv1, which is shown in the middle panel of Fig. 1. Our experiments, however, indicate that directly regressing on reviewer scores provides limited effectiveness. We identify two underlying challenges. First, reviewer confidence tends to be systematically overlooked. Although reviewers vary in their expertise and confidence when evaluating a submission, existing formulations typically treat all scores as equally informative, which reduces the robustness of the supervision signal. Second, review scores lack a unified and stable evaluation standard. Scoring criteria differ across research domains, publication years, and even individual reviewers, introducing inconsistencies that pose challenges for calibrating direct regression.

Building on these observations, we propose NAIPv2 (originating from **N**ewborn **A**rticle **I**mpact **P**rediction), a fast and debiased framework for paper quality assessment. NAIPv2 learns the relative ordering of submissions within the same domain–year group in a pairwise manner, thereby mitigating biases introduced by cross-domain, temporal variations. During inference, the framework is decoupled into a simplified single-branch pointwise regressor, enabling robust score prediction while preserving linear-time complexity. In addition, we introduce a probabilistic treatment of review scores and reviewer confidences, where each score is viewed as a noisy observation of the latent paper quality and confidence determines its uncertainty, yielding a principled comparative signal that guides pairwise ranking. Experimental results demonstrate that our method attains state-of-the-art performance (78.2% AUC and 0.432 Spearman) while offering substantially improved

inference efficiency. Notably, on unseen NeurIPS submissions, the predicted scores align precisely with decision categories, consistently assigning the lowest values to rejected papers and the highest values to oral presentations.

In summary, our main contributions are as follows:

1. **A large-scale domain-labeled dataset.** We release **NAIDv2**, a dataset of 24,276 ICLR submissions (2021–2025) with parsed PDF content and clustered domain labels, enabling large-scale cross-domain debiasing and research on scholarly evaluation.

2. **A scalable and debiased quality estimation framework.** We propose **NAIPv2**, which combines pairwise training with efficient pointwise inference for early-stage paper quality prediction, achieving state-of-the-art performance and strong generalization on unseen NeurIPS review data.

3. **A confidence-aware probabilistic signal.** We introduce **RTS**, a confidence-aware probabilistic formulation that improves the stability of quality estimation.

## 2    RELATED WORK

**Peer Review Dataset.** The blind nature of the peer-review process makes publicly accessible review data extremely limited. Among major conferences, ICLR is particularly notable for its open policy, providing comprehensive review information such as reviewer comments, rating scores, and area-chair decisions, and thus has been used by most existing studies (Staudinger et al., 2024). However, reliance on ICLR also carries substantial risks of information leakage. A common and almost inevitable issue is the inadvertent overlap between training and test sets, where the training data of one method may appear in another method's test set (Zhang et al., 2025a), motivating many works to construct their datasets from scratch. Another source of leakage arises from large language models; for instance, PeerRead (Kang et al., 2018), released in 2018, has likely been incorporated into LLM training corpora, raising concerns about unintended memorization. The training, validation, and test sets of NAIDv2 are constructed following rigorous academic practice. Building on a random split, we ensure that the test set exclusively contains the latest ICLR 2025 submissions, thereby preventing any potential information leakage. In addition, compared with existing datasets, NAIDv2 offers a distinctive advantage by providing multi-granular topic labels along with multimodal PDF-parsed representations.

**Automated Peer Review.** Automated peer-review systems aim to approximate the human reviewing process by taking the full manuscript as input and producing corresponding review reports. To optimize performance on review-related tasks, most automated peer-review methods perform task-specific fine-tuning. Yu et al. (2024) introduces a modular framework for standardization, evaluation, and analysis, and further proposes a mismatch score to measure paper–review alignment. OpenReviewer (Idahl & Ahmadi, 2024) fine-tunes an 8B-parameter LLM on expert reviews to produce structured, guideline-compliant feedback. DeepReview (Zhu et al., 2025) employs a multi-stage pipeline integrating structured analysis, literature retrieval, and evidence-based reasoning. CycleReviewer Weng et al. (2025) goes beyond single-pass reviewing by coupling manuscript generation with iterative peer review through reinforcement learning, aiming to automate the entire research-review cycle. Together, these approaches exploit LLMs' CoT reasoning, boosting performance but incurring substantial computational cost. In contrast to automated peer-review systems, NAIPv2 provides numerical estimates of paper quality for a wide range of AI-for-science applications, enabling fast ranking of candidate papers even when only the title and abstract are available.

**Numerical Quality Estimation.** Predicting the quality of scientific papers has long been an active research area (Zhao & Feng, 2022; Xia et al., 2023; Thelwall, 2025). Traditional approaches typically rely on early impact signals, such as citation counts, author profiles, venue prestige, and linguistic indicators extracted from the manuscript. More recently, researchers have begun to explore LLM-based methods. For example, De et al. (de Winter, 2024) examined whether Chat-GPT can estimate bibliometric indicators (*e.g.*, citation counts), and found that its estimates correlate more strongly with ground-truth impact metrics than traditional readability-based measures. NAIP (Zhao et al., 2025) further used LLMs to directly predict a field- and time-normalized impact score ($TNCSI_{sp}$) from titles and abstracts, achieving an MAE of 0.216. However, prior work (Cortes & Lawrence, 2021; Dougherty & Horne, 2022) has shown that reviewer-annotated

paper quality exhibits little correlation with citation-based impact. Although several studies have attempted to predict metrics derived from peer review (Ghosal et al., 2019; Bharti et al., 2021), they have neither adequately addressed translational inconsistency nor systematically examined the role of reviewer confidence, instead largely overlooking it. In contrast, NAIPv2 learns from the probabilistic aggregation signal RTS and adopts a pairwise learning framework to mitigate translational inconsistency, achieving competitive performance with substantially improved speed.

**Pairwise Ranking.** Pairwise ranking methods have long been established as effective approaches in information retrieval and recommendation (Liu et al., 2009; Cao et al., 2007), yet their adoption in peer review and quality estimation remains limited. Cousins et al. (2025) show that classical pairwise models like Bradley–Terry (Bradley & Terry, 1952) and Thurstone–Mosteller (Thurstone, 1994) can be derived from basic axioms of human comparative judgment. Zhang et al. (2025e) investigate pairwise manuscript evaluation using off-the-shelf LLMs. Owing to the quadratic growth of pairwise comparisons, they demonstrate that aggregating fewer than 2% of all possible pairs is sufficient to recover a robust global ranking. Distinct from prior studies, this work addresses debiasing and efficiency by training in a pairwise manner while performing pointwise inference, thereby achieving inference with linear complexity.

## 3 METHODS

### 3.1 REVIEW TENDENCY SIGNAL (RTS)

In practice, reviewers are not always domain experts in the exact area of a submission, which may prevent them from providing accurate evaluations. To mitigate this source of uncertainty, most conferences and journals require reviewers to report not only an overall score but also an accompanying confidence level, reflecting the degree of trust they place in their own assessment.

Our proposed solution is to treat each review not as a fixed score but as a confidence-weighted estimate of the latent "true quality" of the paper. In this view, the reported overall score $s_i$ reflects a reviewer's central judgment, while the confidence level $c_i$ controls how reliable that judgment is. High-confidence reviews contribute more sharply to the aggregated signal, whereas low-confidence reviews contribute more diffusely, reflecting greater uncertainty.

Formally, we aggregate multiple reviewer scores by treating each score as a confidence-modulated estimate of the latent true quality $x \in [0, 1]$. Rather than relying on fixed weighting heuristics, we employ a smooth Gaussian-shaped scheme in which reviewer confidence directly controls the sharpness of its contribution. After normalizing both scores and confidence levels to $[0, 1]$ (using a lower bound of 0.2 for confidence normalization), each review $i$ contributes a term of the form

$$p(s_i \mid x, c_i) \propto \mathcal{N}\big(s_i \mid x, \sigma(c_i)^2\big), \tag{1}$$

where $\sigma(c_i) = 0.2(1 - c_i) + 0.05$ is a smooth linear mapping that decreases with confidence, assigning greater influence to high-confidence reviews while down-weighting more uncertain ones (see Appendix E for the detailed specification of $\sigma(c_i)$).

Combining $n$ such terms yields a Gaussian-shaped aggregate over $x$:

$$p(x \mid s_{1:n}, c_{1:n}) \propto \prod_{i=1}^{n} p(s_i \mid x, c_i), \tag{2}$$

whose closed-form mean is given by a confidence-adjusted precision weighting:

$$\mathbb{E}[x \mid s_{1:n}, c_{1:n}] = \frac{\sum_{i=1}^{n} s_i \, \sigma(c_i)^{-2}}{\sum_{i=1}^{n} \sigma(c_i)^{-2}}. \tag{3}$$

Because the latent quality is constrained to $[0, 1]$, we take the RTS to be the mean of this aggregate truncated and renormalized over the interval:

$$\text{RTS} = \mathbb{E}[x \mid s_{1:n}, c_{1:n}, \, x \in [0, 1]]. \tag{4}$$

RTS provides a smooth and interpretable mechanism for incorporating reviewer confidence: high-confidence assessments exert greater influence, while low- confidence assessments remain included without dominating. Unlike simple weighted averaging, RTS preserves uncertainty structure in a principled way and produces a final score that is guaranteed to lie within $[0, 1]$. Additional details of RTS are provided in Appendix E.

## 3.2 CONSTRUCTION OF THE NAIDv2 DATASET

To facilitate efficient debiasing in training and evaluation, we construct the NAIDv2 (where **D** denotes **D**ataset). Following established community practices, we build it from ICLR conference review data provided by OpenReview.net, ensuring an acceptance ratio that closely reflects real-world conditions. Given that review data is continually updated, earlier studies either failed to incorporate the latest ICLR reviews or covered a narrower time span (Zhu et al., 2025; Weng et al., 2025). To address this, we developed a dedicated data engine and used it to compile NAIDv2, encompassing 24,276 samples spanning 2021–2025. Although our experiments relied only on review scores, confidence ratings, and related fields, the underlying SQL-based dataset also retains additional information, including the full textual review comments. Furthermore, we downloaded the corresponding original PDF submissions and applied MinerU (Wang et al., 2024a) for structured parsing (additional multimodal experiments are provided in Appendix I).

In addition to its scale and coverage, NAIDv2 is distinguished by its explicit handling of domain bias (as depicted in Fig. 2). As discussed earlier, raw review scores exhibit inconsistencies across domains and across time. To correct for these distortions, we move beyond noisy keyword-based domain labeling and instead adopt a clustering-driven strategy. Specifically, we encode paper titles and abstracts using Qwen3-Embedding-4B (Zhang et al., 2025d) and apply hierarchical clustering (Bar-Joseph et al., 2001; Müllner, 2011) to identify latent domains based on semantic relatedness. Accordingly, we also compute the RTS from the raw scores, providing a robust signal for debiased training. This pipeline produces domain-normalized labels that more faithfully capture expert consensus and enable debiased pairwise training.

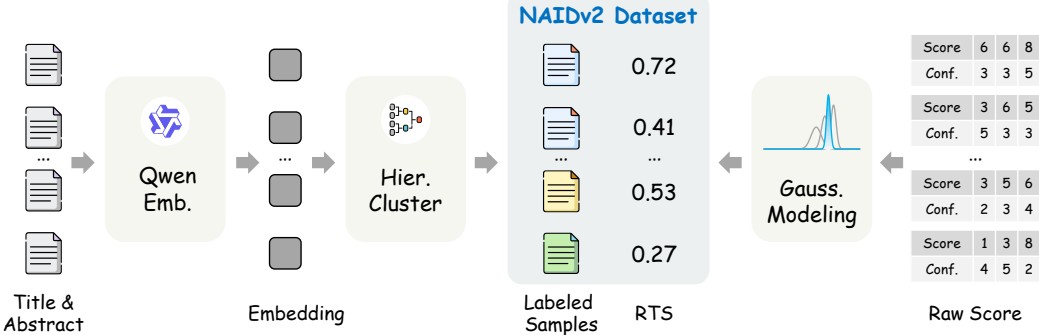

Figure 2: **Debiased data construction in NAIDv2.** Titles and abstracts are embedded and hierarchically clustered into latent domains to mitigate domain bias, while raw scores with reviewer confidences are modeled via Gaussian likelihoods to calculate RTS, yielding a normalized dataset for robust debiased pairwise training.

For additional information on dataset construction, please refer to the Appendix H.

## 3.3 PAIRWISE TRAINING AND POINTWISE SCORING WITH NAIPv2

**Pairwise Training Stage**. NAIPv2 adopts a Bradley–Terry style pairwise learning paradigm, in which the model is trained to capture *relative* differences in RTS between two papers rather than regressing their absolute values (as illustrated in the rightmost panel of Fig. 1). For a pair of submissions $(a, b)$ with ground-truth signals $RTS_a$ and $RTS_b$, we construct a binary preference label $RTS_{ab} = \mathbb{I}[RTS_a > RTS_b]$, where $\mathbb{I}[\cdot]$ denotes the indicator function (equal to 1 if the condition holds and 0 otherwise). Each paper is encoded independently by a shared LLaMA-3 (Dubey et al., 2024) backbone followed by a lightweight MLP head, producing scalar predictions $\hat{y}_a$ and $\hat{y}_b$.

The Bradley–Terry preference probability is then defined as:

$$\hat{z}_{ab} = \text{sigmoid}(\hat{y}_a - \hat{y}_b), \tag{5}$$

where $\text{sigmoid}(\cdot)$ denotes the logistic function. The model is optimized using the standard binary cross-entropy loss shown in Eq. 6, with alternative pairwise objectives discussed in Appendix D.

$$\mathcal{L}(a, b) = -\Big( RTS_{ab} \log \hat{z}_{ab} + (1 - RTS_{ab}) \log(1 - \hat{z}_{ab}) \Big). \tag{6}$$

To mitigate distributional biases across domains and years, training pairs $(a, b)$ are restricted to submissions within the same cluster and the same publication year. This design prevents spurious comparisons caused by cross-field or temporal inconsistencies.

To further enhance robustness and stability, we employ a curriculum learning mechanism based on the gap in $RTS$. Specifically, we partition training pairs into difficulty buckets according to:

$$\Delta_{ab} = |\, RTS_a - RTS_b \,|, \tag{7}$$

where pairs with larger $\Delta_{ab}$ are considered easier, while those with smaller $\Delta_{ab}$ are treated as harder. During the early stages of training, easier pairs dominate the sampling distribution, guiding the network to quickly capture coarse quality distinctions. As training progresses, harder pairs are gradually upsampled, enabling the model to learn fine-grained quality differences.

**Pointwise Inference Stage**. Although training supervision is provided only on submission pairs $(a, b)$, the shared backbone guarantees that each input is independently mapped to a scalar score:

$$\hat{y}_a = f(x_a; \theta), \quad \hat{y}_b = f(x_b; \theta), \tag{8}$$

where $x_a$ and $x_b$ denote the textual representations of submissions $a$ and $b$, respectively. In other words, the pairwise objective implicitly induces a decoupled pointwise scoring function under shared parameters $\theta$.

At inference time, this function naturally generalizes to single-paper evaluation:

$$\hat{y} = f(x; \theta), \tag{9}$$

which provides a pointwise estimate of paper quality that is directly comparable across submissions within the same domain-year subgroup. Although the pairwise loss constrains only relative differences, the shared backbone forces all submissions into a common representation space, thereby yielding in practice a globally consistent scoring scale.

## 4 EXPERIMENTS

### 4.1 METRICS

As discussed in the introduction, NAIPv2 is designed to support a variety of application scenarios. To reflect these use cases, we evaluate the model using two complementary families of metrics. The classification metrics assess performance in settings related to automated decision prediction, allowing direct comparison with related work. The ranking metrics, on the other hand, align with the intended downstream applications of NAIPv2 and focus on performance in tasks such as literature retrieval and recommendation.

For classification, we report Accuracy, F1, ROC-AUC, and Pairwise Accuracy (P. Acc). For ranking-oriented evaluation, we report Spearman's $\rho$ and NDCG@20. Formal definitions of all metrics are provided in Appendix C.

### 4.2 IMPLEMENTATION DETAILS

We divide NAIDv2 into training, validation, and test subsets. The validation set is sampled from the training data with an internal 9:1 split and is used for hyperparameter tuning. To avoid potential information leakage (e.g., earlier ICLR submissions that might have been included in the LLaMA-3 pretraining corpus), the test set is restricted to submissions from the year 2025, comprising 1,029 samples. NAIPv2 is fine-tuned on 4×A40 GPUs (48 GB each) using 8-bit quantization and LoRA adaptation (Hu et al., 2022), with a training time of roughly 1 hour for 10k pairs (constructed from 23,247 samples). On a more common consumer GPU (RTX 3090, 24 GB), the same configuration is conservatively estimated to complete within 6 hours, making the approach accessible to most academic labs. The main hyperparameters include a learning rate of $1 \times 10^{-4}$, a batch size of 8, and training for one epoch (more implementation details are provided in the Supplementary Material).

Table 1: Performance comparison on the ICLR review prediction task. ↑ denotes the higher the better. Results marked with * are taken from Zhu et al. (2025); Yu et al. (2024). "-" means the metric is not reported. Bold numbers denote the best results.

| Category | Method | Acc ↑ | F1 ↑ | AUC ↑ | NDCG ↑ | $\rho$ |
|---|---|---|---|---|---|---|
| - | Lower Bound (Random) | 0.514 | 0.410 | 0.527 | 0.525 | 0.002 |
| | Upper Bound (Info. Leak) | 0.819 | 0.757 | 0.894 | 0.995 | 0.984 |
| API | API-pointwise (ChatGPT) | 0.644 | 0.427 | 0.654 | 0.702 | 0.315 |
| | API-pairwise (ChatGPT) | 0.658 | 0.448 | 0.655 | 0.686 | 0.297 |
| | AgentReview* (Gemini) | 0.424 | 0.424 | - | - | 0.097 |
| | AI Scientist* (Gemini) | 0.614 | 0.481 | - | - | 0.120 |
| AutoRegressive | SEA-EA* (7B) | 0.582 | 0.563 | - | - | - |
| | CycleReviewer* (8B) | 0.678 | 0.559 | - | - | 0.279 |
| | CycleReviewer* (70B) | 0.678 | 0.574 | - | - | 0.267 |
| | DeepReview* (14B) | 0.689 | **0.623** | - | - | 0.405 |
| Regress Only | NAIPv1 (8B) | 0.545 | 0.472 | 0.605 | 0.629 | 0.183 |
| | **NAIPv2** (*ours*, 8B) | **0.706** | 0.609 | **0.782** | **0.771** | **0.432** |

## 4.3 COMPARISON WITH PREVIOUS APPROACHES

As shown in Tab. 1, we compare our proposed method against various approaches. Existing methods can be broadly divided into three categories. Category I consists of prompting LLMs to predict review scores or acceptance decisions based on paper titles, abstracts, or full texts. Owing to the training-free nature of API calls, this paradigm has gained popularity, though its long-term usage costs remain high. Category II encompasses approaches that fine-tune LLMs for automatic reviewing tasks or develop specialized frameworks. The goal of these methods is to maximize the reasoning and analytical strengths of LLMs, enabling them to produce comprehensive review texts. Category III fine-tunes LLMs to regress specific numerical values. Since these approaches avoid the autoregressive generation process, they achieve much faster inference speeds. In addition, we also report the empirical upper and lower bounds on the NAIDv2 Test Set. The upper bound is obtained by training an MLP with cross-entropy loss, where the inputs are reviewer-score statistics (an 11-dimensional vector including the number of reviews and summary statistics of both scores and confidences), and the ground truth is the binary acceptance decision (0/1). The lower bound is determined by a stochastic model.

For the first category, we implement the "API (pointwise)" and "API (pairwise)" approaches with reference to Höpner et al. (2025) (see Appendix F for details). This category has emerged as the most common choice among researchers due to its ease of use; however, neither the pointwise nor the pairwise variant has demonstrated satisfactory performance. The second research direction explores autoregressive inference using LLMs. Due to supervised fine-tuning on specialized data, these methods typically demonstrate stronger performance. DeepReview (Zhu et al., 2025) reaches an F1 score of 62.3 and a Spearman correlation of 0.408 on their test set, demonstrating strong capability on both acceptance decision prediction and score ranking tasks. Nevertheless, since the method generates tokens autoregressively, inference time scales with output length. On mainstream consumer GPUs (e.g., RTX 3090), inference for a single paper can take up to three minutes, which limits its applicability in literature intelligence systems. Regression-only based approaches seek to bypass autoregression by directly predicting a single numerical output. NAIPv1 (Zhao et al., 2025), for example, achieves up to ten predictions per second on the RTX 3090. However, since its training objective is aligned with influence metrics rather than paper quality, its predictive quality is only marginally better than random guessing. By contrast, our proposed approach leverages a debiased pairwise learning design, matching NAIPv1 in inference efficiency (over a thousand times faster than typical autoregressive approaches) while attaining performance on par with DeepReview.

## 4.4 COMPARISON OF LEARNING PARADIGMS AND INFERENCE COMPLEXITY

We further analyze the paradigm differences to underscore the advantages of the NAIPv2 design. Table 2 compares three variants: (i) a pointwise method based on the NAIPv1 architecture but using RTS labels in place of the originals, (ii) a pairwise variant that adopts the same architec-

ture as the pointwise model but concatenates two papers into a single input sequence and predicts a binary 0/1 label, and (iii) our proposed NAIPv2. The pointwise approach suffers from inconsistencies, resulting in suboptimal performance. The concatenation-based variant achieves higher accuracy, but its inference requires explicit pairwise comparisons, leading to a complexity of $O(C \log C)$, where $C$ denotes the number of candidate papers. In contrast, NAIPv2 incorporates debiased pairwise learning during training while retaining pointwise inference, thus preserving the linear complexity $O(C)$ of pointwise methods and achieving superior performance.

## 4.5 EFFECT OF CLUSTERING STRATEGIES ON PERFORMANCE

The strategy chosen for pairwise grouping plays a crucial role in the model's capacity to distinguish between papers within the same set. Table 3 presents the performance differences across grouping methods. When no grouping is applied, or when grouping is based solely on temporal information, the model achieves only marginal improvements over direct prediction, and the results remain below optimal. Grouping via GPT-generated keywords performs worst, with accuracy approaching random guessing. This outcome is mainly attributed to the overabundance of candidate keywords, which produces sparse valid pairs and insufficient training data. In contrast, hierarchical clustering outperforms temporal grouping alone, and the combination of temporal and hierarchical grouping yields the strongest overall results. These findings highlight the critical role of debiasing in pairwise learning for paper quality assessment.

Table 2: Comparison of alternative paradigms. The last column reports the theoretical lower-bound inference complexity of each paradigm.

| Paradigm | AUC | $\rho$ | Th. Compl. |
|---|---|---|---|
| Pointwise | 0.633 | 0.237 | $O(C)$ |
| Pairwise Concat | 0.720 | 0.351 | $O(C \log C)$ |
| NAIPv2 (*ours*) | **0.782** | **0.432** | $O(C)$ |

Figure 3 further examines how the maximum distance parameter (max distance) in hierarchical clustering affects performance. As this parameter becomes smaller, the clustering grows increasingly fine-grained, eventually producing an excessive number of small groups with very few samples, which introduces sparsity and weakens the reliability of pairwise supervision. As the parameter becomes larger, the clustering becomes overly coarse, merging most papers into only a few broad groups and diminishing the benefits of structured comparison. The results show that both extremes are detrimental, and that an intermediate setting provides a more stable and informative partitioning. In particular, the strongest performance is achieved when the max distance parameter is set to 1.0.

Table 3: Impact of Pair Grouping Strategies. P. Acc stands for Pairwise Accuracy.

| Group By | AUC | P. Acc | $\rho$ |
|---|---|---|---|
| None | 0.739 | 0.630 | 0.400 |
| Pub. Time | 0.736 | 0.638 | 0.392 |
| Keyword | 0.556 | 0.620 | 0.007 |
| Hier. Cluster | 0.753 | 0.639 | 0.401 |
| **Time+Hier.** | **0.782** | **0.649** | **0.432** |

Table 4: Superiority of RTS.

| Signal | AUC | P. Acc | $\rho$ |
|---|---|---|---|
| Cites | 0.583 | 0.556 | 0.358 |
| Mean | 0.753 | 0.586 | 0.385 |
| Weighted | 0.754 | 0.598 | 0.402 |
| Median | 0.757 | 0.599 | 0.398 |
| Mode | 0.730 | 0.564 | 0.343 |
| **RTS (ours)** | **0.782** | **0.609** | **0.432** |

## 4.6 BENEFIT OF THE PROPOSED RTS

Table 4 compares RTS with several commonly used quality signals. When citation count (Cites) is adopted as the signal for pairwise training, its AUC performance is substantially lower, partially supporting prior observations by Cortes & Lawrence (2021); Dougherty & Horne (2022). Simple averaging (Mean) and its confidence-weighted variant differ only marginally, indicating that naïvely incorporating confidence provides limited value. Median aggregation performs slightly better, although the overall improvement remains modest. Mode aggregation yields weaker results, as it discards important distributional structure in reviewer scores. In contrast, RTS delivers consistently stronger performance: its probabilistic treatment of reviewer confidence enables a more informative and principled integration of scores, resulting in a more discriminative and reliable quality signal than all traditional signals.

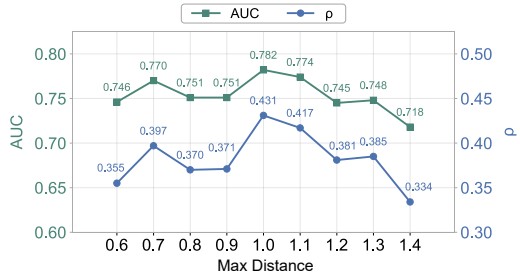

Figure 3: Effect of clustering granularity.

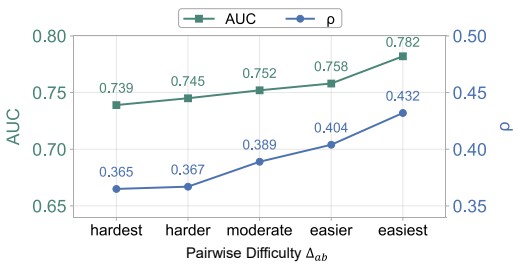

Figure 4: Effect of difficulty bucketing.

### 4.7 LEARNING WITH DIFFICULTY-AWARE SUPERVISION

To study the effect of pairwise difficulty, we group training pairs into buckets by the $RTS$ gap $\Delta_{ab}$ (detailed in Appendix G), where larger gaps indicate easier pairs and smaller gaps correspond to harder ones. Figure 4 shows that model performance improves as the proportion of easy pairs within the training mixture increases. This finding suggests that simple comparisons provide more stable and reliable learning signals, whereas emphasizing hard pairs tends to introduce noise and degrade performance. Building on this observation, we further adopt a curriculum learning strategy. In the early training stages, sampling is biased toward easy pairs, allowing the model to first capture coarse distinctions. Over time, harder pairs are gradually introduced, enabling the model to refine its judgment on subtle quality differences. The results in Tab. 5 verify that curriculum learning provides measurable improvements, supporting it as a useful complement to difficulty-aware supervision.

Table 5: Impact of curriculum learning ("w/o" denotes without, "w/" denotes with).

| Setting | AUC | $\rho$ |
|---|---|---|
| w/o Curriculum | 0.770 | 0.418 |
| w/ Curriculum | 0.782 | 0.432 |

### 4.8 SCALING BEHAVIOR OF NAIPv2

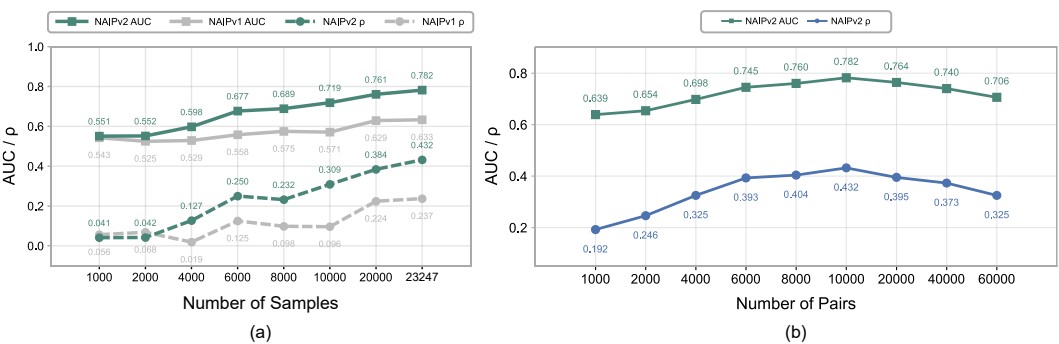

Figure 5: Effect of data size on model performance. (a) Performance comparison of NAIPv1 and NAIPv2 with different numbers of samples, measured by AUC and Spearman's $\rho$. (b) Performance of NAIPv2 with different numbers of training pairs.

We analyze the data efficiency of NAIPv2 by varying both the number of supervised samples and the number of constructed pairs. As shown in Fig. 5, increasing the amount of training samples consistently improves performance, and NAIPv2 maintains an advantage over NAIPv1 across all scales. When varying the number of pairs, NAIPv2 shows steady gains up to roughly 10,000 pairs, after which performance plateaus and may decline slightly, suggesting diminishing returns from adding redundant constraints. Taken together, these trends indicate that the pairwise formulation enables NAIPv2 to extract more effective supervision from a limited set of examples, achieving strong performance even when trained with relatively few samples.

## 5 FROM ICLR TO NEURIPS: A GENERALIZATION CASE STUDY

To further assess the generalization of our model, we conduct a cross-venue evaluation on NeurIPS. Following the same protocol as for NAIDv2, we construct a test set of 13,223 submissions from 2021 to 2024. Unlike ICLR, NeurIPS exhibits an imbalance in review availability: reviews of accepted papers are always public, whereas reviews of rejected papers appear only when authors choose to disclose them. This practice inflates the fraction of accepted papers in the accessible record relative to the true acceptance rate of about 20%, producing a strongly skewed distribution. Such imbalance limits the meaningfulness of prior tasks built on this dataset.

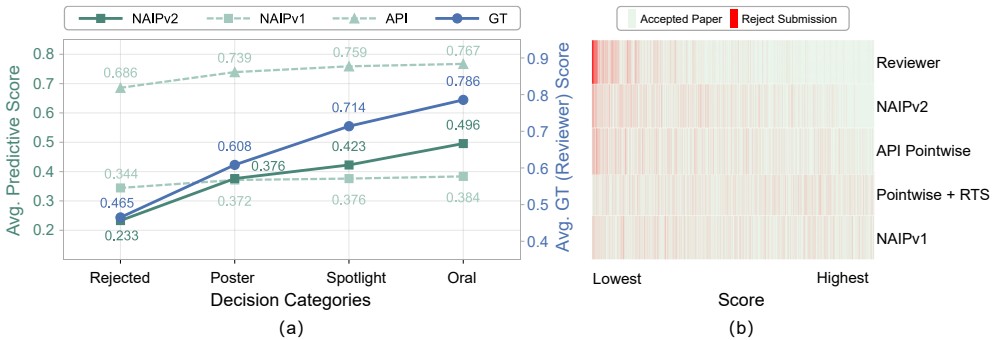

Figure 6: Cross-venue evaluation on NeurIPS. (a) Predicted scores follow the hierarchical structure of conference decisions. (b) Distribution of rejected papers across score ranges.

Given these limitations, we examine a more reliable property of the NeurIPS process: its hierarchical decision structure. NeurIPS organizes accepted papers into posters, spotlights, and orals, forming a quality-ordered sequence above rejected submissions. Under this structure, stronger papers should receive higher evaluation scores and appear in more selective categories. We therefore test whether our predicted scores (logistic-sigmoid normalized) align with this hierarchy. As shown in Fig. 6 (a), our model produces a clear monotonic increase from rejected to poster, spotlight, and oral papers, with a more consistent trend than NAIPv1 and the pointwise API-based method. This alignment indicates that our approach is well calibrated with respect to the observed conference decisions.

In Fig. 6 (b), we further show the distribution of rejected papers across score ranges. Each row corresponds to the min-to-max assigned scores (Green denotes accepted papers, while red denotes rejections). The reviewer row can be regarded as the ground-truth distribution. As expected, most rejections cluster in the lower-score region, yet some appear in higher ranges, reflecting the valuable role of expert preferences in calibrating the review process. NAIPv2 exhibits a trend similar to reviewer assessments, with lower-score regions containing more rejected papers, while the high-score regions contain fewer rejections than the reviewer-assigned scores. The API-based approach shows a weaker tendency, yet still outperforms stochastic prediction. Owing to inconsistencies, NAIPv1 and direct RTS regression perform poorly, as rejected papers are distributed almost randomly across score ranges. This case study shows that NAIPv2 achieves stable generalization and consistent alignment with conference outcomes, suggesting promising directions for broader applications.

## 6 CONCLUSION

This work presents NAIPv2, a debiased and scalable framework for automated paper quality estimation. By employing pairwise learning within domain-year groups and introducing a confidence-aware probabilistic signal, NAIPv2 mitigates score inconsistencies while enabling efficient linear-time inference. On the curated ICLR test set, the model achieves state-of-the-art performance, demonstrating both accuracy and efficiency compared to prior approaches. Moreover, in a cross-venue evaluation on NeurIPS, NAIPv2 predicts a clear monotonic trend from rejected to oral papers and accurately places less rejections in the high-score regions, demonstrating consistency with human judgments and robustness under distributional shifts. Taken together, these results highlight NAIPv2 as an exploratory step toward robust scalable scientific intelligence systems, while leaving open important avenues for future research on quality prediction and broader academic applications.

ACKNOWLEDGMENTS

This research was supported by the Fund of the National Natural Science Foundation of China (Grant No. 62576177), Shenzhen Science and Technology Program (QNXMB20250701090801002, JCYJ20250604184027034, JCYJ20240813114237048), Guangdong Basic and Applied Basic Research Foundation (2026A1515011435), the Fundamental Research Funds for the Central Universities 070-63253222, and the Tianjin Key Laboratory of Visual Computing and Intelligent Perception (VCIP). Computation is supported by the Supercomputing Center of Nankai University (NKSC). "Science and Technology Yongjiang 2035" key technology breakthrough plan project (2025Z053), Chinese government-guided local science and technology development fund projects (scientific and technological achievement transfer and transformation projects) (254Z0102G), National Science Fund of China under Grant No. 62361166670. This work was also supported by the Academy for Advanced Interdisciplinary Studies, Nankai University (AAIS, NKU).

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

# A ETHICAL STATEMENT

The scores presented in this work are generated by machine learning models trained on historical peer-review data and should be understood as black-box predictions, not as objective or authoritative measures of scientific quality. These scores are not intended, nor should they be used, to replace human peer review or editorial decision-making. Our contribution is solely for research purposes, with the aim of studying peer-review dynamics and developing methods for literature intelligence systems. The authors bear no responsibility for any use of the predicted scores in real-world decision processes.

# B DISCLOSURE OF LLM USAGE

ChatGPT was employed during the experimental stage as an auxiliary tool for code debugging, code development, and data visualization. All code was thoroughly reviewed by the authors to ensure accuracy and reliability. ChatGPT also assisted in polishing the language of the manuscript, but was not involved in substantive writing. The authors take full responsibility for any issues arising from the use of ChatGPT.

# C EVALUATION METRICS

This appendix provides the formal definitions of the metrics used in our experiments. We group them into two categories: classification metrics and ranking metrics.

## C.1 CLASSIFICATION METRICS

**ROC-AUC.** Let $P$ and $N$ denote the sets of positive and negative samples. The ROC-AUC for model predictions $\hat{y}$ is defined as

$$\text{AUC} = \frac{1}{|P||N|} \sum_{i \in P} \sum_{j \in N} \mathbb{I}[\hat{y}_i > \hat{y}_j]. \tag{10}$$

**Pairwise Accuracy (PairAcc).** Let $\tilde{y}$ be the thresholded predictions and $\Omega = \{(i,j) : y_i \neq y_j\}$ be the set of cross-label pairs. PairAcc is computed as

$$\text{PairAcc} = \frac{1}{|\Omega|} \sum_{(i,j) \in \Omega} \mathbb{I}[(\tilde{y}_i - \tilde{y}_j)(y_i - y_j) > 0]. \tag{11}$$

**Accuracy and F1.** We follow standard definitions of Accuracy and F1-score.

## C.2 RANKING METRICS

**Spearman's Rank Correlation ($\rho$).** Let $d_i$ denote the difference between the predicted and ground-truth ranks of item $i$. Spearman's $\rho$ is defined as

$$\rho = 1 - \frac{6 \sum_{i=1}^{n} d_i^2}{n(n^2 - 1)}. \tag{12}$$

**NDCG@k.** Let $s_i$ denote the ground-truth relevance of the item at position $i$. The Discounted Cumulative Gain is

$$\text{DCG@}k = \sum_{i=1}^{k} \frac{2^{s_i} - 1}{\log_2(i+1)}. \tag{13}$$

Let IDCG@$k$ be the maximum possible DCG@k. Then

$$\text{NDCG@}k = \frac{\text{DCG@}k}{\text{IDCG@}k}. \tag{14}$$

In our experiments, we use $k = 20$.

## D    DEBIASED PAIRWISE LEARNING OBJECTIVES

In this section we broaden our investigation beyond the main loss and systematically examine alternative objectives for pairwise learning. The motivation is to understand the design space of loss functions and the trade-offs they entail, rather than to commit to a single formulation. Formally, given a pair $(a, b)$ with binary label $z_{ab} = \mathbb{I}[RTSa > RTSb]$, outputs $y_a, y_b$, and difference $d_{ab} = y_a - y_b$, we consider the following representative families.

**(A) Likelihood-based objectives.**    These methods interpret the pairwise preference as a probability that $a$ is better than $b$, and optimize a proper scoring rule between predicted and observed preference.

*Thurstone–Probit.* Replacing the logistic sigmoid with a probit link yields

$$\hat{z}_{ab}^{\mathsf{probit}} = \Phi\left(\tfrac{d_{ab}}{\tau}\right), \quad \mathcal{L}_{\mathsf{Probit}}(a, b) = -\Big(z_{ab} \log \hat{z}_{ab}^{\mathsf{probit}} + (1 - z_{ab}) \log(1 - \hat{z}_{ab}^{\mathsf{probit}})\Big), \qquad (15)$$

where $\Phi(\cdot)$ is the standard normal CDF and $\tau$ controls the slope. This treats preference as arising from a latent Gaussian noise model, offering smoother gradients when comparisons are uncertain.

*Thurstone–Mosteller (TM).* The TM model assumes that each paper $i$ has a latent Gaussian utility with mean $\mu_i$ and variance $\sigma_i^2$. The preference probability is

$$\hat{z}_{ab}^{\mathsf{TM}} = \Phi\left(\frac{\mu_a - \mu_b}{\sqrt{\sigma_a^2 + \sigma_b^2}}\right), \qquad (16)$$

where $\Phi(\cdot)$ is the standard normal CDF. The loss is the same likelihood-based objective,

$$\mathcal{L}_{\mathsf{TM}}(a, b) = -\Big(z_{ab} \log \hat{z}_{ab}^{\mathsf{TM}} + (1 - z_{ab}) \log(1 - \hat{z}_{ab}^{\mathsf{TM}})\Big). \qquad (17)$$

*Pairwise Brier.* Instead of a log loss, we minimize squared error between predicted probability and label:

$$\mathcal{L}_{\mathsf{Brier}}(a, b) = \big(\sigma(d_{ab}) - z_{ab}\big)^2. \qquad (18)$$

This penalizes deviations quadratically and is less harsh than cross-entropy when predictions are confident but wrong.

**(B) Margin- and regression-style objectives.**    These objectives work directly on the score difference $d_{ab}$, either enforcing a margin or softly aligning it with the review gap $\Delta_{ab}$.

*Hinge margin.* With labels $y_{ab} = 2z_{ab} - 1 \in \{-1, +1\}$, we require $d_{ab}$ to exceed a margin $m$:

$$\mathcal{L}_{\mathsf{Hinge}}(a, b) = \big[m - y_{ab}\, d_{ab}\big]_+, \quad [x]_+ = \max(x, 0). \qquad (19)$$

This loss is zero if the preferred item is already ahead by at least $m$, and otherwise grows linearly—encouraging confident separation.

*Huber-on-difference.* Here we compare the predicted difference to the observed review score gap $\Delta_{ab}$. Residual $r_{ab} = d_{ab} - \kappa\, \Delta_{ab}$ is penalized with a robust Huber function:

$$\mathcal{L}_{\mathsf{HuberDiff}}(a, b) = \begin{cases} \frac{1}{2} r_{ab}^2, & |r_{ab}| \le \delta, \\ \delta\big(|r_{ab}| - \frac{\delta}{2}\big), & |r_{ab}| > \delta. \end{cases} \qquad (20)$$

This encourages alignment with the magnitude of human score gaps, while limiting the influence of outliers beyond threshold $\delta$.

**(C) Hybrid objectives.**    These combine a directional likelihood with a light regression regularizer, aiming to preserve ordering while stabilizing score magnitudes.

*Rank-then-Regress.* We augment RankNet with an $L_2$ penalty on the difference–gap residual:

$$\mathcal{L}_{\mathsf{RtR}}(a, b) = -\Big(z_{ab} \log \sigma(d_{ab}) + (1 - z_{ab}) \log(1 - \sigma(d_{ab}))\Big) + \lambda\, (d_{ab} - \kappa\, \Delta_{ab})^2, \qquad (21)$$

Table 6: Comparison of different loss functions for pairwise learning.

| Category | Loss | Acc | F1 | AUC | NDCG | $\rho$ |
|---|---|---|---|---|---|---|
| Likelihood-based | BCE (*adopted*) | 0.706 | **0.609** | **0.782** | 0.771 | **0.432** |
| | Thurstone–Probit | 0.689 | 0.594 | 0.756 | 0.762 | 0.393 |
| | Thurstone–Mosteller | **0.717** | 0.592 | 0.685 | 0.759 | 0.395 |
| | Pairwise Brier | 0.606 | 0.541 | 0.685 | 0.750 | 0.293 |
| Margin / Regression | Hinge Margin | 0.711 | 0.606 | 0.772 | **0.783** | 0.420 |
| | Huber-on-difference | 0.531 | 0.548 | 0.699 | 0.758 | 0.288 |
| Hybrid | Rank-then-Regress | 0.681 | 0.591 | 0.765 | 0.789 | 0.396 |

where $\lambda$ controls the regression weight. This formulation preserves pairwise preference consistency while encouraging smoother calibration within domain–year subgroups.

Overall, results in Tab. 6 support our choice of a purely pairwise likelihood objective as the most stable formulation. Importantly, once regression-style supervision is introduced, performance degrades across metrics, suggesting that absolute review scores suffer from translational inconsistencies across domains and years. This reinforces the view that enforcing numerical alignment is detrimental, whereas ordinal-only objectives provide more robust and transferable signals.

## E  NORMALIZATION IN THE COMPUTATION OF RTS

In computing RTS, we do not use the raw review scores directly. Instead, both the scores and the associated confidence values are normalized. Score normalization follows a standard min–max transformation:

$$\tilde{s}_i = \frac{s_i - \min_{j \in \mathcal{R}} s_j}{\max_{j \in \mathcal{R}} s_j - \min_{j \in \mathcal{R}} s_j}, \tag{22}$$

where $s_i$ denotes the score assigned by reviewer $i$ and $\mathcal{R}$ is the set of reviewers for the submission. This ensures $\tilde{s}_i \in [0, 1]$.

Confidence values are normalized with an adjustable lower bound $\alpha$ to control the effective range. The normalized confidence is defined as

$$\tilde{c}_i = \alpha + (1 - \alpha) \frac{c_i - \min_{j \in \mathcal{R}} c_j}{\max_{j \in \mathcal{R}} c_j - \min_{j \in \mathcal{R}} c_j}, \tag{23}$$

where $c_i$ is the reported confidence of reviewer $i$ and $\tilde{c}_i \in [\alpha, 1]$. The parameter $\alpha \in [0, 1]$ determines the relative position of the lowest observed confidence in the mapped interval.

We conduct an ablation study to evaluate different confidence-normalization schemes and alternative forms of $\sigma(c)$, which governs the variance used in the RTS computation. As summarized in Tab. 7, smoother confidence-to-variance mappings yield consistently better performance. A moderate lower bound, specifically $\alpha = 0.2$, provides the strongest overall results. Both substantially smaller and larger values lead to performance degradation, indicating that restricting the effective confidence range to a balanced level is beneficial for stability.

Table 7: Ablation study on RTS normalization strategies. (a) Different formulations of $\sigma(c)$. (b) Effect of the lower bound $\alpha$ for confidence normalization.

(a) Formulations of $\sigma(c)$

| $\sigma(c)$ | AUC | NDCG | $\rho$ |
|---|---|---|---|
| $\sigma_{\text{aggressive}}$ | 0.766 | 0.775 | 0.391 |
| $\sigma_{\text{smooth}}$ | **0.782** | **0.771** | **0.432** |

$\sigma_{aggressive}(c) = 0.4(1 - c) + 0.1,$
$\sigma_{smooth}(c) = 0.2(1 - c) + 0.05.$

(b) Confidence normalization (lower bound $\alpha$)

| $\alpha$ | AUC | NDCG | $\rho$ |
|---|---|---|---|
| 0.0 | 0.756 | **0.796** | 0.391 |
| 0.2 | **0.782** | 0.771 | **0.432** |
| 0.5 | 0.764 | 0.772 | 0.405 |
| 0.8 | 0.769 | 0.771 | 0.395 |

# F   IMPLEMENTATION OF API-ONLY BASELINES

To ensure fair comparison and to examine the limitations of prompt-only approaches, we re-implemented baseline experiments using GPT-5-mini as an AI reviewer. In the *pointwise* setting, the model was provided with the title and abstract of a single paper and instructed to output a normalized quality score in the range $[0, 1]$. In the *pairwise* setting, two papers were presented simultaneously, and the model was required to judge which paper exhibited higher overall quality. To further evaluate the robustness of prompt design, we constructed two distinct prompts for each setting, as reported in Tab. 8. We then evaluated the outputs using both ranking-based and classification-based metrics. For classification, the F1 score was computed under a decision threshold corresponding to the top-30% predicted scores, following standard practice in acceptance prediction. This setup allows us to quantify the effectiveness of prompt-only methods and highlight their inherent weaknesses when compared with our proposed debiased pairwise learning framework.

Table 8: Performance comparison.

| Category | Prompt | F1 | AUC | NDCG | $\rho$ |
|---|---|---|---|---|---|
| Pointwise | "You are an expert academic paper reviewer.\nEvaluate the quality of a single paper based on title and abstract only.\n\nConsider only:\n- Novelty and significance\n- Clarity of writing\n- Technical soundness (as inferred from abstract)\n\nYou MUST output a STRICT JSON object ONLY with this exact schema:\n{"score":  0.00}\n\nThe score must be a normalized quality between 0 and 1 (inclusive),\nformatted with exactly two decimal places.  Do NOT include any extra keys or text." | 0.427 | 0.654 | 0.702 | 0.315 |
|  | "Review the paper from its title and abstract.\nJudge quality by originality, clarity, and soundness.\n\nReturn ONLY a JSON object:\n{"score":  0.00}\n\nScore ∈ $[0, 1]$, exactly two decimals. No other text." | 0.422 | 0.637 | 0.667 | 0.278 |
| Pairwise | "You are an expert academic paper reviewer.\nYour job is to decide which of two papers is of higher overall quality\nbased on title and abstract only.\n\nConsider only:\n- Novelty and significance\n- Clarity of writing\n- Technical soundness (as inferred from abstract)\n\nYou MUST choose exactly one better paper (no ties).\nOutput a STRICT JSON object ONLY, with this exact schema:\n{"better":  a} OR {"better":  b}\n\nDo NOT include any explanations, notes, or extra fields.\nDo NOT include reasoning.\nDo NOT include any text outside the JSON object." | 0.448 | 0.655 | 0.686 | 0.297 |
|  | "Decide which paper (a or b) is of higher quality\nbased on title and abstract only. Consider novelty, clarity, and\ntechnical soundness.  Pick exactly one.\n\nOutput STRICT JSON:\n{"better":  a} OR {"better":  b}\n\nNo extra text." | 0.439 | 0.659 | 0.690 | 0.316 |

## G    Bucket Distribution Definition

As noted in Section 4.7, we discretize the pairwise $RTS$ gap $\Delta_{ab}$ into buckets with edges $[0, 0.1, 0.2, \ldots, 1.0]$, which correspond to 9 intervals. We define five representative ratio distributions over these buckets to characterize different difficulty settings, ranging from *Easiest* to *Hardest*. Table 9 lists the exact proportions. In particular, easier distributions allocate more weight to large gaps ($> 0.8$), whereas harder ones emphasize small gaps ($< 0.1$). The moderate case is close to uniform and serves as a neutral baseline.

Table 9: Bucket ratio definitions for different difficulty levels.

| **Difficulty** | $< 0.1$ | 0.1–0.2 | 0.2–0.3 | 0.3–0.4 | 0.4–0.5 | 0.5–0.6 | 0.6–0.7 | 0.7–0.8 | $> 0.8$ |
|---|---|---|---|---|---|---|---|---|---|
| Easiest | 0.03 | 0.04 | 0.06 | 0.08 | 0.10 | 0.12 | 0.14 | 0.16 | 0.27 |
| Easier | 0.05 | 0.06 | 0.07 | 0.08 | 0.10 | 0.12 | 0.15 | 0.17 | 0.20 |
| Moderate | 0.111 | 0.111 | 0.111 | 0.111 | 0.111 | 0.111 | 0.111 | 0.111 | 0.112 |
| Harder | 0.20 | 0.17 | 0.15 | 0.13 | 0.10 | 0.08 | 0.07 | 0.06 | 0.04 |
| Hardest | 0.27 | 0.16 | 0.14 | 0.12 | 0.10 | 0.08 | 0.06 | 0.04 | 0.03 |

## H    Data Engine for Constructing NAIDv2

To enable scalable data collection and management, we implemented a dedicated data engine for NAIDv2 (Tab. 10). The engine adopts a three-layer architecture: the Controller layer interfaces with external APIs (e.g., OpenReview) and orchestrates data flow; the Service layer parses raw submissions and reviews, normalizes metadata (such as confidence scores), and exports processed datasets; and the DAO layer provides object-oriented access to the underlying MySQL database.

The database consists of two core tables, 'Papers' and 'Reviews', linked by the paper identifier. The 'Papers' table stores submission-level metadata (title, abstract, venue, derived GT labels, etc.), while the 'Reviews' table stores reviewer comments, ratings, and rebuttals. Automatic crawlers retrieve all submissions from OpenReview, insert missing entries, and populate reviews through structured parsing.

This modular design ensures reproducibility, facilitates large-scale updates across multiple years of OpenReview data, and supports downstream model training and evaluation.

## I    Effect of Additional Information

As noted in NAIPv1 (Zhao et al., 2025), incorporating additional information can sometimes enhance model performance, though such gains are not always consistent. In this study, adhering to the double-blind requirement, we extracted auxiliary information directly from the papers and incorporated it into network training. Specifically, we employed a computer vision-based approach to extract key figures and key tables: the scores for the key figures were derived through network inference, and the table descriptions were generated using analysis by GPT-5-mini. However, our experiments show that adding such auxiliary content does not improve performance across the tested settings. In fact, models augmented with introductions, conclusions, figure scores, or table descriptions all perform worse than those relying solely on titles and abstracts. These results are broadly consistent with prior findings (Zhou et al., 2024), indicating that not all types of supplementary content provide useful signals for review score prediction, further confirming that abstracts already contain sufficient information.

## J    Additional Remarks on Experimental Comparisons

In Tab. 1, we compare the performance of existing methods, reporting some results directly from the original publications. We were unable to reproduce SEA-E (Yu et al., 2024), OpenReviewer (Idahl & Ahmadi, 2024), DeepReview Zhu et al. (2025), and CycleReviewer (Weng et al., 2025) for two primary reasons. First, the training and test sets used in these works differ and sometimes overlap, raising the possibility that our test data appear in their training sets (and vice versa). Second,

Table 10: Schema of the Papers and Reviews tables in NAIDv2 (partial view). PK = Primary Key, FK = Foreign Key.

(a) Papers Table

| Field | Type | Desc. |
|---|---|---|
| id | BIGINT | PK |
| title | VARCHAR(255) | Title |
| abstract | TEXT | Abstract |
| external_id | VARCHAR(255) | Openreview ID |
| openreview_venue | VARCHAR(255) | Venue |
| scores | JSON | Review scores |
| confs | JSON | Confidences |
| pub_year | INT | Year |
| embedding | JSON | Embedding vec |
| cluster_cat | VARCHAR(64) | Cluster label |

(b) Reviews Table

| Field | Type | Desc. |
|---|---|---|
| id | BIGINT | PK |
| paper_id | BIGINT | FK → Papers.id |
| belong_to | VARCHAR(128) | Group ID |
| reply_to | VARCHAR(128) | Reply target |
| conf_score | FLOAT | Confidence |
| rec_score | FLOAT | Recommendation |
| reply_title | TEXT | Reply title |
| review_summary | TEXT | Summary |
| strengths | TEXT | Strengths |
| weakness | TEXT | Weaknesses |

Table 11: Effect of additional information on paper quality estimation.

| Method | AUC | NDCG | $\rho$ |
|---|---|---|---|
| Title & Abstract (*ours*) | **0.782** | 0.771 | **0.432** |
| +Introdution | 0.716 | 0.760 | 0.321 |
| +Conclusion | 0.750 | 0.768 | 0.395 |
| +Key Figure Score | 0.769 | **0.798** | 0.417 |
| +Key Table Description | 0.758 | 0.753 | 0.381 |

retraining these models is computationally infeasible. For example, CycleReviewer requires training a 123B-parameter model, which far exceeds our available resources. Moreover, we note that instances from benchmark datasets may be embedded in the training data of these methods, suggesting that their reported gains could partly reflect data leakage rather than genuine improvements in generalization.

Nonetheless, most prior studies follow a relatively consistent dataset construction protocol, typically sampling randomly from the ICLR dataset. As shown in Fig. 7, the distribution of our test set is well aligned with that of DeepReview, with a Jensen–Shannon of 0.079. This alignment indicates that, despite the aforementioned limitations, the comparisons in Tab. 1 remain reasonably meaningful across methods.

To further validate the effectiveness of NAIPv2, we retrain our model on the DeepReview-13K training set and evaluate it on the corresponding test set. Despite relying on a smaller backbone (8B vs. 14B), NAIPv2 delivers competitive, and in several cases superior, performance, underscoring the strength of our debiased pairwise learning framework. We additionally advocate for a title-based data-splitting protocol, which enables researchers to determine whether a paper belongs to the training or test set using only its title.

| Method | 2024 F1 ↑ | 2024 P.Acc ↑ | 2024 $\rho$ | 2025 F1 ↑ | 2025 P.Acc ↑ | 2025 $\rho$ |
|---|---|---|---|---|---|---|
| DeepReview (14B) | 63.07 | 62.42 | **35.59** | **62.27** | **64.02** | 40.47 |
| NAIPv2 (ours, 8B) | **66.13** | **63.37** | 33.22 | 58.94 | 60.87 | **40.82** |

Table 12: Results on the DeepReview-13K dataset (Zhu et al., 2025), with all hyperparameters kept identical to those used in the main paper.

## K  FURTHER DETAILS ON EMBEDDING AND CLUSTERING

A key motivation for our embedding–clustering design arises from the observation that review scores can vary substantially across domains. However, directly applying keyword-based normalization often suffers from issues of uncontrollable granularity and drifting category boundaries, which weaken its reliability. For instance, research on remote sensing (Li et al., 2024a) may be regarded either as satellite-based sensing (Ali et al., 2025; Kuckreja et al., 2024; Zhang et al., 2025c) or as visual perception (Girshick, 2015; Li et al., 2025a; Zhang et al., 2025b; Li et al., 2026; **?**). "PromptKD" (Li et al., 2024b) might be situated within prompt learning (Li et al., 2024c; Wu et al., 2024; Chen et al.,

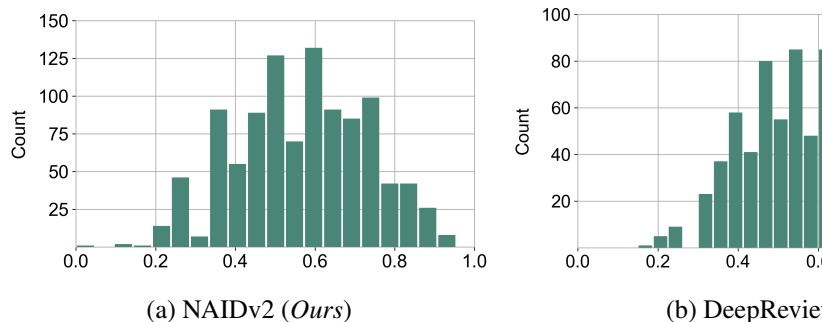

(a) NAIDv2 (*Ours*)                    (b) DeepReview

Figure 7: **Data distribution comparison between NAIDv2 (ours) and DeepReview Test 2025 datasets.** The two distributions are well aligned, with a Jensen–Shannon of 0.079, indicating comparability of evaluation results across methods.

2024b) or knowledge distillation (Hinton et al., 2015; Li et al., 2023). To address this challenge, we employ Qwen3-Embedding in combination with hierarchical clustering, which enables semantically grounded grouping of papers beyond brittle keyword matching.

As shown in Table 13 and Table 14, this setup allows us to systematically study the impact of embedding and clustering choices. Both BGE-M3 (Chen et al., 2024a) and Qwen3-Embedding provide strong baselines, yet Qwen3-Embedding paired with hierarchical clustering achieves the most consistent gains. HDBSCAN McInnes et al. (2017) remains competitive but trails slightly in AUC and $\rho$. Importantly, embeddings enriched with explicit clustering-oriented instructions outperform those without, and incorporating both titles and abstracts yields better representations than titles alone. Among the prompts we explored, the one explicitly directing the model to group semantically similar papers provides the most balanced improvements across all evaluation metrics.

Table 13: Impact of embedding models and clustering algorithms on retrieval performance. (a) Comparison of embedding models. (b) Comparison of clustering algorithms with confidence normalization.

(a) Embedding models

| Model | AUC↑ | NDCG↑ | $\rho$ |
|---|---|---|---|
| BGE-M3 | 0.774 | 0.769 | 0.417 |
| Qwen3-emb | **0.782** | **0.771** | **0.432** |

(b) Clustering algorithms

| Method | AUC↑ | NDCG↑ | $\rho$ |
|---|---|---|---|
| HDBSCAN | 0.765 | **0.803** | 0.422 |
| Hier. Cluster | **0.782** | 0.771 | **0.432** |

Table 14: Comparison of different instruction prompt designs

| Instruction | AUC↑ | NDCG↑ | $\rho$ |
|---|---|---|---|
| None *(no instruction)* | 0.759 | 0.775 | 0.266 |
| Identify the main category of scholar papers based on the titles and abstracts | 0.767 | **0.831** | 0.416 |
| Given a title of a scientific paper, retrieve the titles of other relevant papers | 0.768 | 0.784 | 0.412 |
| Cluster similar papers as close as possible based on title and abstract (*ours*) | **0.782** | 0.771 | **0.432** |

## L    ABLATION ON PAIRWISE SAMPLING STRATEGIES

To further investigate the robustness of the proposed pairwise learning strategy, we conduct ablation studies by introducing additional constraints during pair construction. The first experiment (Table 15 Panel (a)) examines the effect of limiting the maximum number of times a sample can appear in training pairs. The results show that imposing such restrictions does not provide clear benefits. In fact, the best overall performance is achieved when no limitation is enforced, suggesting that

allowing samples to be paired freely provides greater diversity and stronger supervision signals for training.

The second experiment (Table 15 Panel (b)) analyzes the role of enforcing a minimum score difference between paired samples. Without such a margin, performance degrades, indicating that constructing pairs from nearly indistinguishable items introduces noise into the training signal. A margin of 0.05 provides the best overall balance, improving both AUC and Spearman correlation, while larger margins ($\geq 0.1$) result in performance drops due to reduced pair availability. These findings highlight the importance of carefully controlling pair selection to enhance both the reliability and effectiveness of pairwise learning.

Table 15: Effect of pairwise learning constraints. (a) Influence of limiting the maximum occurrence times of each sample in pairwise training. (b) Influence of enforcing a minimum score difference when constructing pairs.

(a) Maximum occurrence times

| Setting | AUC | NDCG | $\rho$ |
|---|---|---|---|
| 2 | 0.776 | 0.773 | 0.426 |
| 4 | 0.766 | 0.791 | 0.408 |
| 8 | 0.767 | 0.774 | 0.391 |
| 16 | 0.782 | 0.771 | 0.432 |
| 32 | **0.782** | **0.771** | **0.432** |

(b) Minimum score difference

| Setting | AUC | NDCG | $\rho$ |
|---|---|---|---|
| 0 | 0.752 | 0.778 | 0.395 |
| 0.05 | **0.782** | 0.771 | **0.432** |
| 0.1 | 0.753 | **0.785** | 0.397 |
| 0.15 | 0.770 | 0.778 | 0.402 |
| 0.2 | 0.761 | 0.774 | 0.404 |

## M  CROSS-ARCHITECTURE EVALUATION OF NAIPv2

We further evaluate the transferability of NAIPv2 on two additional LLM families: *Qwen* (Bai et al., 2023) and *Mistral* (Jiang et al., 2023). These experiments aim to verify whether the gains brought by our debiased pairwise learning framework are specific to a single backbone or generalize across architectures.

The results in Table 16 demonstrate that the core idea of NAIPv2 continues to yield consistent improvements on both Qwen and Mistral models. In both cases, replacing pointwise regression with our pairwise training strategy leads to substantial gains across all evaluation metrics, including accuracy, F1, AUC, NDCG, and Spearman correlation. These findings indicate that NAIPv2 is *not* tied to a particular LLM family and transfers well across different backbone architectures.

Table 16: Performance comparison on Qwen and Mistral backbones. Pairwise training consistently improves evaluation metrics across both model families.

| Approach | Acc ↑ | F1 ↑ | AUC ↑ | NDCG ↑ | Spearman |
|---|---|---|---|---|---|
| Qwen-7B Pointwise | 0.555 | 0.521 | 0.656 | 0.682 | 0.232 |
| Qwen-7B Pairwise | **0.653** | **0.576** | **0.738** | **0.786** | **0.357** |
| Mistral-8B Pointwise | 0.567 | 0.536 | 0.675 | 0.769 | 0.285 |
| Mistral-8B Pairwise | **0.621** | **0.553** | **0.711** | **0.827** | **0.342** |

## N  DISCUSSION AND PRACTICAL CONSIDERATIONS

As shown in Tab. 17, NAIPv1 performs strongly in predicting the academic impact of newly published articles, whereas NAIPv2 underperforms on the NAIDv1 benchmark. This result appears consistent with the observation of Cortes & Lawrence (2021) that impact and quality are often only weakly correlated.

These findings suggest that there is no single "universal key" for all tasks, and the choice of method should depend on the specific application. API-based solutions are easy to use and broadly accessible, but their performance is limited (Tab. 1); fine-tuned autoregressive models benefit from domain-specific optimization and achieve

Table 17: Performance of NAIPv1/v2 on article impact prediction task.

| Model | NDCG | $\rho$ |
|---|---|---|
| NAIPv1 | **0.901** | **0.459** |
| NAIPv2 | 0.761 | 0.279 |

stronger results, though at the cost of additional deploy-
ment requirements; NAIPv2 provides performance com-
parable to these autoregressive approaches with much faster inference, but lacks the interpretive
outputs that autoregressive models offer.

Overall, these comparisons indicate that no single method prevails across all scenarios, underscoring
the importance for both research and industry communities to recognize and leverage the comple-
mentary strengths of these approaches.

