# OpenReview forum: "NAIPv2: Debiased Pairwise Learning for Efficient Paper Quality Estimation"
_ICLR.cc/2026/Conference — ICLR 2026 Poster_

### Official Review · Reviewer_qFdP · 2025-10-20

**Soundness:** 3
**Presentation:** 3
**Contribution:** 3
**Rating:** 8
**Confidence:** 3

**Summary:**

LLM-assisted peer review has become a popular topic in recent years, and Paper Quality Estimation is an important part of it. This paper proposes NAIPv2, a model that takes the titles and abstracts of two papers as input and predicts their pairwise quality. The model is based on the Llama 3 backbone, trained on the ICLR dataset with human reviewer’s feedback as ground truth. NAIPv2 achieves state-of-the-art performance (AUC and rank correlation) than several baselines, including autoregressive models. In addition, the method runs in linear time, generalizes well, and works when only abstracts are available instead of full papers.

**Strengths:**

* The paper proposes a paper quality predictor NAIPv2 that achieves state-of-the-art performance using only titles and abstracts, showing a high rank correlation with human reviews.
* NAIPv2 is efficient (linear time), generalizes well, and does not require full text, which makes it practically useful. I look forward to seeing downstream applications, such as helping authors improve their titles and abstracts.
* The paper combines existing tools into a clear and effective workflow. Although it uses Llama 3 as a backbone, given that paper quality estimation is still a new topic, I find the overall novelty acceptable.
* The experiments are well done. Some additional comments are listed in the “Questions” section.

**Weaknesses:**

* I would like to see experiments using more proxies for paper quality (see “Questions” section).
* Section 3.3 is a bit confusing. the training details could be explained more clearly in the main text.

**Questions:**

* Many studies in this area use citation count as a proxy for paper quality because (1) their methods are unsupervised, and (2) human reviews are highly noisy, making it hard to align with them. I suggest adding experiments that use citation-based proxies and comparing them with existing methods, although they are not reliable. Such results could reveal deeper insights: if NAIPv2 performs worse on predicting citations but aligns much better with human reviews, it may suggest that human judgments rely heavily on information from titles and abstracts.
* This is more of a curiosity than a critique: since judging paper quality solely from titles and abstracts is unrealistic, I believe that NAIPv2’s strong alignment with human reviewers may come from certain “cheap signals” in abstract / title. I wonder what those signals might be; perhaps the authors could offer some hypotheses or discussion.

---

> ### Author Response · Authors · 2025-11-19
> **Authors' Response to Reviewer qFdP**
>
> We sincerely thank the reviewer for the high recognition of our work’s **novelty, practical value, and methodological clarity**. We appreciate the constructive feedback and address the concerns below.
>
>
> ## Clarity of Training Details
>
> We apologize for any confusion caused by the current presentation. We will revise Section 3.3 to more clearly describe the training and implementation details.
>
> ## Suggestion on Treating Citation Count as Proxy Signal
>
> We acknowledge the reviewer’s suggestion and understand the motivation. Our hesitation in using citation-based proxies mainly comes from three considerations:
>
> 1. **Severe cross-domain variability**: citation levels vary drastically across fields—for example, even among top-tier venues, CA (oncology top tier journal) has an impact factor more than twenty times that of TPAMI—indicating that cross-field differences often overshadow the signal from individual paper quality.
>
> 2. **Community discouraging citation-based evaluation**: major scientific assessment frameworks such as **DORA** and the **Leiden Manifesto** explicitly discourage the use of citation metrics for evaluating individual research outputs.
>
> 3. **Weak empirical correlation**: prior analysis by Cortes (NeruIPS Board Member) [1] reports an extremely weak correlation between citation counts and review scores (Pearson ≈ 0.051), suggesting very limited utility as a reliable proxy for human judgments.
>
> Despite these concerns, we still conducted experiments using citation counts as a proxy. The results are shown below:
>
>
> | Signal     | AUC ↑  | P. Acc ↑ | ρ     |
> | - | - | - | - |
> | Mean  | 0.753 | 0.586  | 0.385 |
> | Mode | 0.730 | 0.564  | 0.343 |
> | Cites   |   0.583    | 0.556 |   0.358    |
> | RTS (ours) | 0.782 | 0.609  | 0.432 |
>
> The results show that using citation count as a proxy signal performs poorly, and further indicate that it **should not** be used for pairwise training. However, we hesitate to interpret this as evidence that *“human judgments rely heavily on information from titles and abstracts”*, because:
>
> - To collect citation data, we queried Semantic Scholar using paper titles and found that **9,441 papers were not indexed at all**, causing the effective dataset size to shrink by nearly **one-third**.
> - Early studies [2,3] have already shown that citation-related metrics can be predicted from the title and abstract alone. Subsequent research [4-6] has also indicated that providing additional information does not always translate into improved overall performance.
>
> We believe that human reviewers rely on far more than just the title and abstract when forming their assessments. Their decisions reflect a holistic understanding of the entire paper, involving reasoning patterns that are extremely difficult to model and value judgements that may never be fully captured in a mathematical framework. This is also one of the key reasons why we oppose using NAIPv2’s output scores directly in the human peer-review process.
>
> We will include this experiment in the revised version.
>
>
> ## Curiosity about Cheap Signals
> Regarding "cheap signals", we believe that titles and abstracts often serve as a compressed reflection of both **the scientific quality of the work** and **the academic maturity of the author team**. Strong papers typically present:
>
> - clear background and motivation,
> - a novel methodological contribution, and
> - leading empirical results,
>
> all of which are summarized explicitly in the abstract. Additionally, the quality of title selection and abstract writing often reflects the authors’ overall scholarly rigor.
>
> Our view is that the LLM primarily captures **patterns learned from large-scale scientific text**, rather than relying on superficial cues.
>
>
> We once again thank you for the insightful suggestions and positive assessment, and we will incorporate the corresponding improvements into the revised manuscript.
>
> ### References
> [1] Cortes, Corinna, and Neil D. Lawrence. "Inconsistency in conference peer review: Revisiting the 2014 neurips experiment." arXiv preprint arXiv:2109.09774 (2021).
>
> [2] de Winter, Joost. "Can ChatGPT be used to predict citation counts, readership, and social media interaction? An exploration among 2222 scientific abstracts." Scientometrics 129.4 (2024): 2469-2487.
>
> [3] Zhao, Penghai, et al. "From Words to Worth: Newborn Article Impact Prediction with LLM." Proceedings of the AAAI Conference on Artificial Intelligence. Vol. 39. No. 1. 2025.
>
> [4] Höpner, Niklas, et al. "Automatic Evaluation Metrics for Artificially Generated Scientific Research." arXiv preprint arXiv:2503.05712 (2025).
>
> [5] Zhou, Ruiyang, Lu Chen, and Kai Yu. "Is LLM a reliable reviewer? a comprehensive evaluation of LLM on automatic paper reviewing tasks." LREC-COLING 2024
>
> [6] Vital Jr, Adilson, et al. "Predicting citation impact of research papers using GPT and other text embeddings." Physica A: Statistical Mechanics and its Applications (2025): 130789.

---

> > ### Comment · Reviewer_qFdP · 2025-11-20
> > **Reviewer Response**
> >
> > The author's response resolved my questions and requests and I am satisfied with it. While I believe the paper has some minor issues, its highlights, especially the metric is independent from the full text, make it worthy of publication in ICLR. I will maintain my current evaluation.
> >
> > Furthermore, I read the comments from other reviewers and agree with the weaknesses mentioned by Reviewer g7oZ. However, I did not read the author's response to Reviewer g7oZ in detail (it was too long, and I am not an expert in this area). I believe Reviewer g7oZ's satisfaction with the author's response is crucial.

---

> > > ### Author Response · Authors · 2025-11-21
> > > **Authors' Appreciation to Reviewer qFdP**
> > >
> > > We sincerely thank you once again for your supportive assessment and valuable feedback. We are glad that our responses addressed your concerns.

---

### Official Review · Reviewer_64Pf · 2025-10-31

**Soundness:** 3
**Presentation:** 3
**Contribution:** 2
**Rating:** 2
**Confidence:** 4

**Summary:**

As the number of submissions to peer-reviewed conferences increases exponentially each year, systems that automatically assign a review score to submissions may become increasingly important. This paper proposes to train an LLM to assign a score to a research article solely based on its title and abstract.
It first introduces a new dataset, NAIPD2, curated using  ICLR submissions from 2021 to 2025.
To account for reviewer confidence, it proposes a confidence-normalised target score, RTS, which is used as the ground-truth label for each paper. The paper claims that reviewing style (and hence review scores) varies by the paper’s domain and year. To account for this bias, instead of directly regressing on the RTS score, they propose a pairwise loss computed using a siamese-style network over the pairs selected from the same domain and year. The network is trained via BCE loss to predict which paper in a training pair has a higher RTS score.  To create domain buckets, instead of using the given keywords, the paper proposes to use hierarchical clustering of the paper’s title and abstract embeddings. To ensure that the LLMs being trained haven’t seen the papers/reviews before, it proposes to use only 2025 submissions in the test set.
Thorough ablations justify the design choices.

**Strengths:**

1. The paper is clearly written, with most of the details present either in the main paper or the appendix.

2. The paper has thorough ablations to justify all the design choices.

3. The paper releases a new dataset that may help the research community.

**Weaknesses:**

1. While I understand that regression-based methods are obviously much more efficient, the lack of any reasoning provided by the model significantly hampers their applicability. On the other hand, methods like DeepReview provided a detailed output that can assist in actually reviewing a paper and perform as good as the proposed method.


2. Another cited contribution is release of Naidv2 dataset. However, similar datasets ( and recipes to create them) already exist, e.g. the dataset used in DeepReview.

3. Another issue is a fair comparison with other baselines. All the reported experiments are ablations for the design choices made by the paper, e.g. pairwise loss vs pointwise loss, curriculum vs no curriculum etc., I don’t see an apples-to-apples comparison with other trained auto-regressive models that may be more inefficient but more accurate and useful.

4. A simple baseline would be an LLM finetuned to generate the review along with the score. This way, all the reviews from the paper can be considered as a separate training sample.

5. Line 322:323: Table 1 reports numbers directly from DeepReview and CycleReview papers, which are computed on different test data (though from ICLR 2025 only). Typically, numbers reported in the same table should be computed on exactly the same dataset.

6. OpenReviewer is mentioned in the related works, but not used as a baseline.

7. The designed system takes only the title and abstract into account, whereas human reviewers would seldom base their decision solely on these two attributes. Does it mean that the system is capturing spurious correlations?

8. While the application of a pairwise loss function to the problem at hand may be novel, the loss function used is the standard one employed to train reward models under Bradley-Terry (BT) reward model assumptions.

**Questions:**

1. Line 247: 248: One of the claims in the paper is that reviewing style varies by domain and year, and clustering is used to create domain buckets. Can you show that review scores indeed follow a different distribution in each cluster-year group? Maybe a plot with cluster-year on the x-axis and some statistic quantifying the distribution on y-axis?

2. Line 213-214 -- “pipeline produces domain-normalized labels”. How is RTS normalised using the domain information?  In eqn 14 and 15, what is j iterating over? Scores of the same paper, or all the scores in the dataset?

3. Given that papers from the same year are used to select the pairs during training, does it mean that we can only compare scores of the papers in the same domain and year? Can we use it to meaningfully compare papers across years?

4. Line 314: 316: Could you please explain the upper bound? Is the input to the MLP just a scalar or a vector of all the review scores for a paper? What if a paper has 3 reviews and another has 4 reviews?

5. To compute F1 and accuracy, you need to select a threshold for the score based on validation data. Is the threshold selection dependent on domain / year?

6. Table 4 / Section 4.6: When using a different statistic (e.g., mean), I hope that it is used both for training and while testing and reporting the numbers in Table 4 - i.e. please clarify that you are not training using mean but then using  RTS to compute ground-truth in pairwise accuracy and ground-truth rank in correlation.

7. Section 4.7: In Figure 4, it is not clear what you are comparing.  You have pairwise difficulty on the x-axis. For a given bucket, say the “easiest” bucket, are you reporting the performance obtained when you train your model using only the “easiest” pairs?  If yes, then how come it matches exactly with the performance in  ‘w/ Curriculum’ setting in Table 5 where you are using all pairs but in a curriculum.



**Comments:**

1. Given that the paper claims RTS as one of its contributions, its details should be moved from the appendix to the main paper.

2. Table 1: Isn’t a higher correlation better? There is no upward arrow sign beside it.

---

> ### Author Response · Authors · 2025-11-19
> **Authors' Response to Reviewer 64Pf - Part 1/3**
>
> We sincerely appreciate the time and effort you invested in evaluating our submission, and recognizing the **clarity of writing**, **completeness of ablations**, and **usefulness of the dataset**. Below we would like to clarify several misunderstandings and respond to each point raised.
>
> Before addressing itemized concerns, we would like to clarify a potenial misconception regarding the **goal and application scope** of our work.
>
> ## **NAIPv2 is NOT intended to serve as an automated reviewing method**
>
> As stated in the Introduction (lines 33–38), NAIPv2 is NOT designed to perform automatic reviewing.
> Our objective is to build an efficient and reliable quality estimation component for large-scale scientific workflows where human reading capacity is severely limited under massive publication volume. Such workflows include AI Scientist systems [1], automated survey pipelines [2], literature search engines [3], and large-scale recommendation systems (cold start stage) [4].
>
> We use peer-review data for supervision because citation-based statistics do not consistently reflect human preference [5]. The choice of supervision signal does not change the intended application of the model. The experiments are designed to show that NAIPv2 can achieve performance comparable to previous SOTA automatic-review approaches only on the shared subtask of quality estimation, rather than performing the review-generation task itself.
>
> Our goal is fundamentally different:
>
> **We aim to build an efficient, bias-reduced quality estimator usable in applications where:**
> - **PDFs cannot be reliably or legally downloaded** (common in literature-search systems, commercial bibliographic tools, and many non-arXiv publishers);
> - **Real-time latency matters** (e.g., ranking hundreds of retrieved documents in search/recommendation/AI-scientist/automatic-survey systems);
> - **Computational resources are limited and cost sensitivity is essential**, requiring lightweight inference rather than full-document LLM processing.
>
> Moreover, in real-world settings **PDFs are not always accessible**, whereas titles and abstracts are consistently and legally available, which makes **our design both practical and necessary**. Our contribution is not intended to compete with automatic reviewing systems, but to provide a complementary and efficient quality-estimation component that supports downstream applications under realistic constraints.
>
>
> ## Prospects for Application
>
> As discussed before, **our method is not meant as a replacement** for such systems.
> Rather, it is **complementary**: Reasoning-rich outputs are useful for human-facing reviewing, but in our intended setting (retrieval/recommendation/AI-scientist pipelines), such outputs significantly increase latency and cost  (*mentioned in Appendix N*).
>
>
> As emphasized in the General Response, our method has broader value beyond the peer-review task itself, such as providing literature intelligence for emerging AI-for-Science systems, serving as a cold-start engines for recommendation pipelines, and offering a fast, even online, reward signal in reinforcement learning. We also believe the core idea introduced in this work may also benefit peer-review modeling directly: by attaching a lightweight regression head to existing autoregressive models and applying pairwise training, one could inject de-biased supervision into model parameters without materially increasing training cost or interfering with language-modeling behavior. This may lead to more reliable quality estimation, ranking, and related capabilities.
>
>
>
> ## Novelty about the Cruated Dataset NAIDv2
>
> Due to the fact that ICLR only provides topic keywords for a very small number of years, **NAIDv2 is, to the best of our knowledge, the only review dataset that offers domain labels**. This work presents not only a dataset but also a complete account of the automatic domain-labeling procedure. All experimental details and results are reported in Appendix L, Tables 12 and 13. We consider this indirect yet systematic approach a meaningful source of novelty when compared with existing datasets.
>
> Moreover, **NAIDv2 is designed as a multimodal dataset** for large-model research. Beyond raw PDFs, it includes parsed JSON structures (processed by minerU [6], Sec. 3.2, line 205-206), extracted figures and tables, and curated core visual elements identified by a specialized detector (Appendix J, lines 911–913).
>
> We believe that this richness will meaningfully support and accelerate progress in the relevant research communities.

---

> ### Author Response · Authors · 2025-11-19
> **Authors' Response to Reviewer 64Pf - Part 2/3**
>
> ## Concern about Fair Comparision
>
> Regarding raised weakness 3-5, we want to clarify that this challenge is essentially unavoidable for nearly all research in this area. Most early approaches rely almost entirely on ICLR review data, yet they adopt different train–test splits. This creates an inherent structural issue: the test set of method A may appear in the training set of method B, and vice versa. As a result, perfectly controlled comparisons across papers are, in practice, impossible to achieve.
>
> In this context, we have done our best to ensure that our experiments and evaluations remain as fair as possible. Specifically, we examine the distributional similarity between the NAIDv2 test set and the DeepReview-13 2025 test set and find a Jensen–Shannon divergence of 0.079, indicating that the two datasets have highly similar score distributions and thus remain meaningfully comparable (see Appendix K for details).
>
> Nevertheless, to thoroughly address your concern, we **applied and downloaded the official DeepReview-13K dataset** and strictly followed their provided train/test split. We retrained our model under the same setting of NAIPv2, and report the results below.
>
>
> | Method | 2024 F1 ↑| 2024 P.Acc ↑ | 2024 ρ |  2025 F1 ↑ | 2025 P.ACC ↑| 2025 ρ |
> |-------|--------------|----------|--------|--------------|----------|--------|
> | DeepReview (14B) | 63.07| 62.42 | **35.59** | **62.27** | **64.02** | 40.47 |
> | NAIPv2 (ours, 8B) | **66.13** |  **63.37** | 33.22 | 58.94 | 60.87 | **40.82** |
>
> Despite using a **smaller backbone (8B vs. 14B)**, our method still achieves competitive performance, demonstrating the effectiveness of NAIPv2.
>
> We advocate a title-based data-splitting protocol, which would allow any researcher to determine whether a paper belongs to the training or test set solely from its title. Although we believe this is a reasonable direction for the community, it falls outside the scope of the present work.
>
>
> We would also like to reiterate that our work is not intended to compete with autoregressive reviewing systems. As stated in Appendix N (lines 1044-1047), "no single method prevails across all scenarios" and "both research and industry communities should recognize and leverage the complementary strengths of these approaches."
>
> ## Missing OpenReivewer in Tab. 1
> We are unable to compare OpenReviewer with the other methods in Table 1 for two reasons. First, the OpenReviewer paper reports only metrics such as Error Rate. Second, we cannot evaluate it on the NAIDv2 test set because its training data is drawn from ICLR 2022–2025, which overlaps with the sampling range of our test set (containing only 2025 submissions). This issue is common in this research area and falls outside the scope of what this paper aims to address. We call on the research community to establish title-based rules for train/test splits to ensure fair, non-overlapping, and reproducible evaluations.
>
> ## Concern about Using Title & Abstract as Input
> The design choice is dictated by real application constraints:
>
> - PDF access is legally restricted for many venues.
> - Metadata (title & abstract) is universally accessible.
> - Many large-scale applications refrain from parsing full PDF.
>
> Besides, in Appendix J (Table 11), we further include experiments that incorporate multiple types of information extracted from PDFs. The results show no clear improvement, which is consistent with prior findings [7-9].
>
> This suggests that title & abstract already encode rich signals such as motivation, methodology, results, clarity, and even author expertise.
>
>
>
> ## Novel Framing but Conventional Loss
> To the best of our knowledge, **NAIPv2 is the first successful attempt** to integrate a pairwise training strategy into this line of work. Precisely for this reason, we aimed for theoretical elegance and methodological simplicity, prioritizing clarity and extensibility so that future work can readily build upon it. We hope that this design choice will help support and stimulate further research within the community.
>
>
> ## Visualization of Data Distribution
> We provide the visualization on our anonymous project page: **[https://sway.cloud.microsoft/evGxX5hFrlfOjLKa]** (see the rightmost panel).
> The plots reveal clear distributional shifts across both years and cluster-year groups, showing that mean scores and **score profiles vary systematically**.
> This indicates that **reviewing style is not uniform but differs across both domain and year**, providing further empirical support for the motivation behind our method.
>
> By the way, if you are interested in the overall distribution of ICLR submission data, I recommend referring to [10].
>
> ## Ambiguity of line 213-214
> Thank you for pointing out this ambiguity.
> **RTS itself is not domain-normalized**; the effect comes from pariwise training using clustering-derived domain labels.
> We will clarify: in Equations (14–15), index j iterates over the reviewers of the same paper.

---

> ### Author Response · Authors · 2025-11-19
> **Authors' Response to Reviewer 64Pf - Part 3/3**
>
> ## Comparision across Years/Domains
> Domain–year pairs are trained on a shared backbone, which provides a certain degree of cross-domain comparability. Table 3 shows that grouping by domain alone already yields strong results, suggesting that cross-year comparisons remain meaningful.
>
> Regarding cross domain comparision, we would like to emphasize that such **cross-domain comparisons are rare in practical applications**. Most real-world scenarios involve retrieving papers for a specific topic or keyword, where NAIPv2 typically receives a list of papers returned by the API within the same domain. This naturally aligns with the way the model is trained and allows NAIPv2 to achieve its optimal performance.
>
>
>
>
> ## "Upper bound" and MLP input dimensionality
> The MLP receives an **11-dimensional statistical vector**, including:
>
> - number of reviews
> - mean, std, min, max, median of review scores
> - mean, std, min, max, median of reviewer confidence
>
> Hence, 3 vs. 4 reviews do **NOT** change the input dimensionality.
>
>
> ## Threshold selection for F1/accuracy
> We **did NOT assign separate thresholds for each domain** because we believe that **using a unified threshold is a fairer practice in academic evaluation**. Although adopting domain-specific thresholds could potentially yield higher performance, we avoided this to maintain consistency and comparability across domains. The thresholding rule is described in detail in the main text (line 809).
>
>
> ## Consistency in Table 4 statistics
> We clarify that the statistic used to supervise training is **exactly the same** one used to compute the evaluation metrics. In the **supplementary material**, we provide the complete training and testing code, where the computation of all metrics is explicitly controlled by the gt_field argument in the parser. Under this code setup, the inconsistency you mentioned cannot occur.
>
> ## Question about Difficulty buckets
> We apologize for this confusion. The buckets correspond to global difficulty groupings defined in Appendix H, Tab. 9. They do **NOT** correspond to training on partial subsets. We will clarify this in the main text.
>
> ## Suggestion about Moving RTS Details to the Main Paper
> We will move more details of RTS into the main paper. Thank you.
>
> ## A Potential Missing Upward Arrow
> As ρ ranges from –1 to 1, values near 0 already indicate poor performance.
> Therefore, adding an upward arrow would be **misleading**.
>
> We appreciate your contribution to improving the quality of our manuscript, and we fully understand the earlier misunderstanding while respecting your original assessment. We would greatly appreciate it if you could kindly reconsider your evaluation in light of the clarifications we have provided.
>
> ### References
>
> [1] Lu, Chris, et al. "The ai scientist: Towards fully automated open-ended scientific discovery." arXiv preprint arXiv:2408.06292 (2024).
>
> [2] Wang, Yidong, et al. "Autosurvey: Large language models can automatically write surveys." Advances in neural information processing systems 37 (2024): 115119-115145.
>
> [3] He, Yichen, et al. "Pasa: An llm agent for comprehensive academic paper search." arXiv preprint arXiv:2501.10120 (2025).
>
> [4] Xu, Zhelin, Shuhei Yamamoto, and Hideo Joho. "Research Paper Recommender System by Considering Users' Information Seeking Behaviors." arXiv preprint arXiv:2504.02377 (2025).
>
> [5] Cortes, Corinna, and Neil D. Lawrence. "Inconsistency in conference peer review: Revisiting the 2014 neurips experiment." arXiv preprint arXiv:2109.09774 (2021).
>
> [6] Wang, Bin, et al. "Mineru: An open-source solution for precise document content extraction." arXiv preprint arXiv:2409.18839 (2024).
>
> [7] Zhao, Penghai, et al. "From Words to Worth: Newborn Article Impact Prediction with LLM." Proceedings of the AAAI Conference on Artificial Intelligence. Vol. 39. No. 1. 2025.
>
> [8] Höpner, Niklas, et al. "Automatic Evaluation Metrics for Artificially Generated Scientific Research." arXiv preprint arXiv:2503.05712 (2025).
>
> [9] Zhou, Ruiyang, Lu Chen, and Kai Yu. "Is LLM a reliable reviewer? a comprehensive evaluation of LLM on automatic paper reviewing tasks." Proceedings of the 2024 joint international conference on computational linguistics, language resources and evaluation (LREC-COLING 2024). 2024.
>
> [10] Yang, Jing, Qiyao Wei, and Jiaxin Pei. "Paper Copilot: Tracking the Evolution of Peer Review in AI Conferences." arXiv preprint arXiv:2510.13201 (2025).

---

### Official Review · Reviewer_f3tL · 2025-10-31

**Soundness:** 4
**Presentation:** 3
**Contribution:** 3
**Rating:** 8
**Confidence:** 3

**Summary:**

This paper introduces NAIPv2, a fast regression model for paper review score estimation. The key idea behind this model is Review Tendency Signal (RTS), which uses not only the raw scores of each paper review but also the specified confidence level to model uncertainty behind individual scores. Alongside the model, the authors also introduce NAIDv2, a collection of ICLR review data compiled from OpenReview.

**Strengths:**

- Clear benefits of using confidence information and RTS over raw scores, as seen in Table 4
- Computational efficiency of NAIPv2 with pointwise inference
- NAIDv2, a collection of reviews and confidence scores from ICLR, appears to be thoroughly constructed.
- Performance gains over more computationally expensive API and autoregressive baselines
- Extensive evaluation of NAIPv2/NAIDv2, including interesting demonstration of model generalizability based on NeurIPS decision category hierarachy

**Weaknesses:**

- It looks like the paper does not exactly mention how regression head was implemented on top of LLaMA 3 for NAIPv2. I didn’t have time to check out NAIPv1 paper so I’m not sure if the architecture was discussed there, but still I think it would be nice to have the architecture description somewhere in the paper.
- It would have been nice if NAIPv2 was also tested on the LLMs other than LLaMA 3 to see if the benefits of NAIPv2 still holds.

**Questions:**

- Regarding Section 3.2, it’s not very intuitive to me whether NAIDv2 really handles domain bias. NAIDv2 only contains ICLR submissions, which seems to be a relatively homogenous set given that ICLR is a ML conference. Are the authors suggesting that two papers of similar content and quality could have significantly different scores just because the authors chose different submission categories?

---

> ### Author Response · Authors · 2025-11-19
> **Authors' Response to Reviewer f3tL**
>
> We thank you for the positive assessment, especially the recognition of the **usefulness of our confidence-aware RTS design**, the **efficiency of NAIPv2**, and the **strength of our extensive experimental evaluation**.
>
> ## Regarding the architecture description
> Thank you for pointing this out. The regression head is a simple MLP without any unique architectural design. In the revision, we will include a concise description of the regression head used on top of LLaMA-3 in NAIPv2 so that the architecture is fully self-contained and does not require referring back to NAIPv1.
>
> ## Regarding evaluation on additional model families
> Following your suggestion, we additionally evaluated NAIPv2 on *Qwen* [1] and *Mistral* [2] models. The results show that the core idea of NAIPv2 continues to yield consistent improvements on Qwen and Mistral models, indicating that our approach is not tied to a single LLM family and transfers well across different architectures.
> | Model                 | Acc ↑  | F1 ↑   | AUC ↑  | NDCG ↑ | Spearman |
> |-|-|-|-|-|-|
> | Qwen-7B Pointwise       | 0.555 | 0.521 | 0.656 | 0.682 | 0.232     |
> | Qwen-7B Pairwise        | 0.653 | 0.576 | 0.738 | 0.786 | 0.357     |
> | Mistral-8B Pointwise  | 0.567 | 0.536 | 0.675 | 0.769 | 0.285     |
> | Mistral-8B Pairwise   | 0.621 | 0.553 | 0.711 | 0.827 | 0.342     |
>
>
> ## Regarding Section 3.2 and the interpretation of “domain bias”
> By “domain bias,” we do *NOT* refer to broad cross-disciplinary differences (e.g., ML vs. biology), but rather to within-ICLR topical and reviewer-pool differences, which can still produce systematic shifts in scoring behavior (check the rightmost pannel of our
> anonymous project page: https://sway.cloud.microsoft/evGxX5hFrlfOjLKa). Different ICLR subject areas, such as theory, optimization, computer vision, and reinforcement learning, tend to be evaluated by partially distinct reviewer pools with different expectations regarding rigor, novelty, and evidentiary standards. As a result, two papers of similar quality may indeed receive different absolute scores depending on which topical area and reviewer pool they are assigned to. Our pairwise formulation and normalization procedures are designed to reduce the influence of such scoring tendencies, enabling the model to focus more on underlying paper quality rather than area-specific scoring conventions.
>
> We thank you again for the constructive suggestions and strong overall assessment. We will incorporate all clarifications and additional experiments in the revised version.
>
> ### References
>
> [1] Bai, Jinze, et al. "Qwen technical report." arXiv preprint arXiv:2309.16609 (2023).
>
> [2] Mistral AI. Ministral-8B-Instruct-2410. huggingface.co/mistralai/Ministral-8B-Instruct-2410.

---

### Official Review · Reviewer_g7oZ · 2025-10-31

**Soundness:** 2
**Presentation:** 2
**Contribution:** 3
**Rating:** 4
**Confidence:** 4

**Summary:**

This paper proposes NAIPv2, a framework for estimating scientific paper quality by combining pairwise learning with a probabilistic Review Tendency Signal (RTS) that integrates reviewer scores and confidences. The authors introduce a new large-scale dataset (NAIDv2) derived from ICLR submissions and report strong results in both within-domain and cross-domain evaluation. The central claim is that NAIPv2 provides debiased, scalable, and efficient quality estimation compared to prior LLM-based or regression-based approaches.
The paper is clearly written and tackles an important and timely problem in automated peer review and scientometric prediction. However, the theoretical framing and probabilistic modeling choices raise several concerns, especially regarding the internal consistency of the modeling assumptions and the clarity of related work.

**Strengths:**

This is an interesting and ambitious paper tackling a key problem in automated peer review. The dataset and efficiency contributions are also substantial.
Equation (1) is well-motivated and welcome to see; it provides a principled probabilistic link between confidence and uncertainty rather than relying on ad hoc weighting.


The proposal to integrate reviewer confidence directly into the score aggregation model addresses a genuine issue in automated peer review and quality estimation.


The focus on efficiency and domain-year stratified training is well thought out and practically motivated.

**Weaknesses:**

(Main critique)
You introduce and motivate a Gaussian probabilistic model (RTS) but then discard it during pairwise ranking. When training with a logistic loss, you implicitly assume Gumbel noise, not Gaussian, effectively switching from a Thurstone–Mosteller to a Bradley–Terry model.


This inconsistency weakens the theoretical story: RTS models uncertainty via Gaussian variance, yet the pairwise comparisons ignore that structure and the confidence information is lost.


For example, if paper A’s reviewers are all high-confidence and paper B’s are all low-confidence, this distinction disappears once RTS is computed. A unified probabilistic model that preserves confidence across comparisons would be both cleaner and more principled.

For the avoidance of doubt, the model to which I refer is essentially a Thurstone-Mosteller model, where we take the probability of one or more reviewers scoring paper A over paper B, given the observed scores we have from reviewers for these papers, using your theoretical model of latent-factor based reviewer-paper scoring, and confidence values as additional inputs.

See e.g.,
[Towards Cognitively-Faithful Decision-Making Models to Improve AI Alignment
Cyrus Cousins, Vijay Keswani, Vincent Conitzer, Hoda Heidari, Jana Schaich Borg, Walter Sinnott-Armstrong] for discussion of probabilistic pairwise prediction models in the broader ML context.

The Thurstone-probit of Appendix D approaches this idea, but variance information has already been removed by this point. I’d like to see this further explored.

Conceptual and Modeling Issues
(186–213)
Many technical aspects are vague in their description, hampering understanding and readability. This is likely a writing quality issue, rather than a criticism of the work behind it.
The introduction of the Review Tendency Signal (RTS) as a posterior mean implicitly assumes a Bayesian framework, but the paper never defines a prior or explains how the posterior is derived. As written, the likelihood $p(s_i \mid x, c_i)$ is defined, but this does not imply a likelihood on $x$. Please clarify whether a uniform prior is assumed, and present the closed-form expression for the posterior mean.


Given your confidence mapping functions, the posterior mean likely admits a clean closed form, which would help motivate the simplicity of your model.


(Appendix F; σ(cᵢ))
The mapping from reviewer confidence to variance is extremely limited. Although you normalize per reviewer, a learned nonlinear function over the entire reviewer population would be both straightforward and far more flexible.


(269)
Clarify that the additive offset correction may depend on year and domain; a global offset is only justified if all pairwise comparisons form a connected component.


(242, 682–696)
The term sigmoid is too vague — you almost certainly mean the logistic function. This matters because later in the appendix you replace it with a Gaussian CDF, which is also a sigmoid; the text should explicitly reference this distinction.


The model should be described more clearly in the language of Bradley–Terry and Thurstone–Mosteller frameworks, which would provide a unifying probabilistic interpretation and clarify the noise assumptions.


The appendix (lines 682–696) is interesting but should be integrated into the main text. The theoretical implications of alternative link functions (logistic vs. Gaussian CDF) should be discussed explicitly, as should the choice of Brier loss. A connection to strictly proper scoring rules would strengthen this section.


Terminology and Framing
The term scale inconsistency appears repeatedly but seems misapplied. The issue appears to be translational inconsistency (offsets), not multiplicative scale differences. The current phrasing is consistent but confusing.

**Questions:**

Related Work and Fairness of Discussion
(124)
The criticism of Zhang et al. (2025a) on the Re² dataset seems unfair and unsubstantiated. The statement “deviations between reported results and those presented in the original papers” needs concrete evidence or examples. If the concern is data leakage or overlap, that should be clearly documented and cited. Otherwise, this reads as an unsupported claim, can the authors clarify?


(154)
The summary of Höpner et al. (2025) lacks essential details. The claim that pairwise ranking is “unreliable” for review scores is unconvincing.


If regression on review scores is feasible, pairwise regression should also be; at worst it merely doubles the noise variance of independent samples.
The critical question is how pairs are selected and what is predicted (rank order, score difference, or ordinal classification). It seems to me that pairwise comparisons are most interesting for ranking, rather than cardinal scores, as you have the additional benefit of simplifying the problem domain to a binary prediction. In cardinal (regression) settings, all that's happening is canceling out translational offsets, which can be interesting when you're controlling for specific effects, but seems unimportant for random pairings. It's not clear from the descriptions in related work which of these methods are being employed, and this seems extremely important. Pairwise methods aren't monolithic: the advantage or lack thereof in using them comes from how pears are chosen and what function of the pair one tries to predict, and I would like to see this explored or discussed.

Broader Modeling Questions:

In automated paper review methods discussed, it seems like there are at least two axes of comparison here.

1: The methods used to make predictions, and 2: what is being predicted. Why not treat citation counts, reviewers scores, and pairwise predictions as separate target, try to predict them all with one model, and select some combination of them for the actual quality estimate? Moreover, these various targets don't seem incompatible. Some concepts are better predictors of quality than others, but likely in complementary ways. It seems a joint predictive model would benefit from shared structure, and this multimodal learning would also make better use of available data.

It would be valuable to separate and jointly analyze multiple predictive targets (e.g., citation counts, reviewer scores, and pairwise preferences) within a unified model. These signals are not incompatible and could benefit from shared structure via multitask learning.



Bias:
It seems to me that, under a Bradley Terry or Thurstone-Mosteller model, you are controlling for bias by subtracting out factors. You subtract out year-dependent and domain dependent factors with this strategy.

The “de-biasing” strategy seems roughly equivalent to including additive terms for each year-domain combination in a regression model, since you're using pairwise comparisons to control for these effects. It would help to compare these approaches empirically or at least discuss their equivalence.

In the same vein, why not restrict pairwise comparisons to papers reviewed by the same reviewers? Although this reduces the number of pairs, it would directly control for reviewer-level bias and may yield cleaner comparisons. It also may alleviate or reduce the need to normalize for each reviewer.

---

> ### Author Response · Authors · 2025-11-19
> **Authors' Response to Reviewer g7oZ - Part 1/3**
>
> We would like to thank you for your thoughtful and constructive feedback, and for recognizing our work as **tackling the important problem**. We appreciate their positive remarks on our **probabilistic formulation linking confidence and uncertainty**, the **explicit use of reviewer confidence in score aggregation**, and the **dataset and efficiency contributions**.
>
>
> ## Consistency between RTS and Pairwise Modeling
> We sincerely thank you for this insightful observation. Following your suggestion, we conducted an additional experiment that more explicitly implements the *Thurstone–Mosteller (TM)* comparison.
>
> Instead of using RTS as a single fused score, we followed your proposal to **separate the mean $\mu$ and the variance $\sigma^2$**, treating:
> - the mean score as the latent quality estimate, and
> - the variance as the uncertainty term entering the TM likelihood.
>
> Concretely, for a pair of papers (A, B), we compute the TM probability under the assumption that their scores are independent：
>
> $P(A>B)=\Phi\left(\frac{\mu_A - \mu_B}{\sqrt{\sigma_A^2 + \sigma_B^2}}\right).$
>
> We re-trained the model using this TM loss and compared it against the adopted Bradley–Terry (BT) formulation. The results are shown below.
>
> | Method | AUC ↑ | NDCG ↑ | Spearman |
> |--------|-----------|--------|-------------|
> | BT (ours) | **0.782** | 0.771 | **0.432** |
> | TM (reviewer suggestion) | 0.759 | **0.783** | 0.395 |
>
> While we initially expected explicit uncertainty modeling to provide additional benefits, the TM-style model performs slightly worse under identical settings. A likely explanation is that additional information does not necessarily lead to better performance, as evidenced in Tables 6 and 11. Our findings suggest that RTS is particularly effective as a variance-aware aggregation mechanism, while BT remains empirically more robust and stable for large-scale pairwise training.
>
> In fact, when we envisioned this work at the beginning, the very first modeling direction we explored was a Thurstone-style approach (as you mentioned). Although we hoped it might deliver stronger results, it consistently underperformed the BT model on the validation set, and its loss exhibited slightly higher variability. These observations were somewhat unexpected, but they illustrate an important aspect of methodological research: well-motivated models do not necessarily translate into the most reliable empirical behavior, and careful experimentation is essential to understanding where different formulations succeed or fall short. This process ultimately clarified the landscape for us and led to the conclusion that the RTS combined with the BT formulation offers the most stable and effective solution among the alternatives we investigated.
>
> We will include this comparison and discussion in the revised manuscript. Although the additional exploration did not lead to performance improvements, we sincerely appreciate your suggestion, which prompted a much deeper analysis. In addition, we consider Cyrus Cousins’ *Towards Cognitively-Faithful Decision-Making Models to Improve AI Alignment* to be an insightful piece of work, and we will discuss it in the revised manuscript.
>
> ## Conceptual Explanation Refinement
> We apologize for the confusion. We assume a non-informative uniform prior. Under this prior, the untruncated posterior is Gaussian with closed-form mean:
> $$
> \mu_{\text{post}}
> =\frac{\sum_i s_i/\sigma(c_i)^2}{\sum_i 1/\sigma(c_i)^2}.
> $$
>
> In the revised paper, we will explicitly state the prior, clarify the Bayesian assumptions, and refine the derivation of the posterior mean. We will also expand the RTS section to improve clarity.
>
> ## Possible Nonlinear σ(c) Extension
> We appreciate this suggestion and agree that a learned mapping could provide greater flexibility. We will consider this direction in future work.
>
> ## Clarifications on Offset Correction
> We thank you for the helpful suggestion. We will clarify the discussion of offset correction in the revised version, including how it relates to different years or domains and when a global formulation is appropriate.
>
> In addition, we would like to note that **cross-domain comparisons are unlikely to arise in the real application scenarios** we target. In automatic literature review (AutoSurvey [1]), for example, the system begins with a keyword-based query, and services such as Semantic Scholar return papers that are already tightly centered around that topic. **NAIPv2 was purposefully crafted with this workflow** in mind, and our training setup was carefully engineered to replicate this very pattern of use. This deliberate design choice is precisely what allows the model to **realize its strongest performance**.

---

> ### Author Response · Authors · 2025-11-19
> **Authors' Response to Reviewer g7oZ - Part 2/3**
>
> ## BT/TM Interpretation Clarification
> We will revise the modeling section to better contextualize our approach with respect to the BT/TM paradigms and to clarify the associated assumptions in the revision.
>
> ## Improving Paper Structure Suggestions
> Thank you for the suggestion. In the revised PDF, we will make full use of the additional page to incorporate your suggestions.
>
> ## Sigmoid Terminology Clarification
> Thank you for pointing this out. We will revise the text to explicitly distinguish the logistic function from the Gaussian CDF in both the main paper and the appendix.
>
> ## Scale Inconsistency Terminology Clarification
> Thank you for this clarification. We will update the terminology carefully.
>
> ## Clarification on Related Work
> Regarding Zhang et al. (2025a), our intention was not to imply misreporting. Our remark on “deviations” refers to the evaluation setup described in Appendix B of Zhang et al., where the authors state that they *“further evaluate some models.”* This suggests that peer-review baselines were **not retrained on Re²**, but instead evaluated using released checkpoints. Because both the Re² dataset and these baselines’ training corpora originate from **OpenReview**, the Re² test set may appear in—or be highly similar to—their training data. This unverifiable overlap helps explain several **unusual performance shifts** relative to the baselines’ original papers. For instance, DeepReviewer and CycleReviewer differ by **<10 F1** in their original work but differ by **28.2** on Re²; similarly, SEA-E, originally the weakest model, unexpectedly achieves the **highest F1** on Re². We will revise the wording to emphasize that our point concerns **evaluation-setting differences**.
>
> Regarding Höpner et al. (2025), our summary aimed to reflect the empirical patterns reported by Höpner et al., not to claim that pairwise ranking is inherently unreliable. In their experiments (Table 2, Figure 3, Table 3), under identical conditions, the **pairwise formulation** performs **worse** than the **direct regression** formulation. This gap likely stems from their specific implementation choices: pair sampling is random, and score aggregation uses a simple averaging strategy, which may underutilize available information. Our intention was to contextualize their results without overgeneralizing. We will revise the text accordingly.
>
> We agree that pairwise approaches are not monolithic, and that the way pairs are sampled and the choice of prediction targets can lead to meaningfully different behaviors. While prior work has explored pairwise ideas conceptually, there is, to our knowledge, no established pairwise training paradigm in this domain; we are the first to conduct end-to-end pairwise training for article quality estimation. Therefore, given that very few studies to date implement genuine pairwise training, the current body of work is insufficient to form a complete or meaningful taxonomy. We will clarify this and revise the relevant statements accordingly based on your suggestions.
>
> ## Discussion about Multitask Modeling
> We agree this is a promising direction. However, we deliberately avoid using citation counts as a primary proxy for “paper quality” due to several limitations:
> (1) citation distributions vary drastically across fields (e.g., Medical top-tier journal CA’s impact factor (=total cites/total publication numbers) is over twenty times that of TPAMI), making cross-field normalization challenging;
> (2) assessment frameworks such as **DORA** and the **Leiden Manifesto** explicitly discourage citation-based evaluation of individual research outputs;
> (3) Analyses by Corinna Cortes *(NeurIPS Board Member)* [2] show extremely weak correlation between citations and review scores (Pearson ≈ 0.051).
>
> Nevertheless, following Reviewer qFdP’s suggestion, we added experiments using citation counts as a proxy, and—as expected—performance is substantially weaker than using RTS. While multitask modeling is intriguing, principled incorporation of citation signals requires careful domain normalization and is beyond the present scope.

---

> ### Author Response · Authors · 2025-11-19
> **Authors' Response to Reviewer g7oZ - Part 3/3**
>
> ## Discussion of “de-biasing” Strategy and Additive Terms
> We acknowledge that, in an idealized setting where scoring scales are consistent and biases are strictly additive, pairwise differencing and pointwise models with year–domain fixed effects could be mathematically equivalent. In our data, however, this **equivalence may not hold**. Pairwise training relies on score differences, which align with how reviewers make comparative judgments and remain stable even when scoring conventions or reviewer scoring ranges vary across years and domains. A pointwise formulation with explicit year–domain parameters assumes that these effects are constant and purely additive, conditions that are not satisfied in ICLR settings. For example, ICLR’s scoring rubric changes across years: in 2021 a score of 4 corresponded to *“Ok but not good enough – rejection,”* whereas this year, a score of 4 corresponds to *“marginally below the acceptance threshold, would not mind if the paper is accepted.”* These inconsistencies further explain why **pointwise formulations underperform** in our setting.
>
>
> ## Clarification on Reviewer-Level Pairwise Restrictions
> While restricting pairwise comparisons to papers reviewed by the same reviewer could control reviewer-specific bias, **reviewer identities are NOT available** in double-blind settings. Therefore, constructing such reviewer-specific pairwise sets is **infeasible in practice**.
>
> We would like to once again thank you for your thoughtful and in-depth review. We hope that our responses, the additional experimental results, and the actions we will take in the revision can adequately address your concerns. We look forward to any further discussion you may have.
>
> ### References
> [1] Wang, Yidong, et al. "Autosurvey: Large language models can automatically write surveys." Advances in neural information processing systems 37 (2024): 115119-115145.
>
> [2] Cortes, Corinna, and Neil D. Lawrence. "Inconsistency in conference peer review: Revisiting the 2014 neurips experiment." arXiv preprint arXiv:2109.09774 (2021).

---

> > ### Comment · Reviewer_g7oZ · 2025-11-23
> >
> > Thank you for the detailed response.
> > I think the TM comparison is valuable, especially for the insight it provides into the modeling, even if the empirical performance is worse. I also think the higher variance isn't too surprising, as it's much more sensitive to outliers than the logistic models, which may be worth investigating as a possible cause. Regardless, this coupled with the presentational changes to the paper discussed above have me optimistic, and I am raising my score to 6.

---

> > > ### Author Response · Authors · 2025-11-24
> > > **Authors' Appreciation to Reviewer g7oZ**
> > >
> > > We are pleased to hear that the clarifications have addressed your concerns and have made your view of this article more positive and satisfactory. Once again, we sincerely appreciate your insightful feedback and the valuable contributions you have made to improving the quality of this manuscript.

---

### Official Review · Reviewer_hKKi · 2025-11-04

**Soundness:** 2
**Presentation:** 2
**Contribution:** 2
**Rating:** 2
**Confidence:** 3

**Summary:**

This paper constructs a dataset (NAIDv2) along with a framework (NAIPv2) for automated paper quality estimation. In particular, the framework adopts pairwise learning (comparing two papers at a time) but is deployed as an efficient pointwise predictor (judging a single paper's quality). The framework also uses a gaussian model, named Review Tendency Signal, to aggregate multiple reviews into a single ground-truth quality signal. Tested on unseen NeurIPS submissions, NAIPv2 is able to achieve better performance over the previous NAIPv1 framework.

**Strengths:**

1. This paper tackles an important problem to collect a massive dataset of paper review and propose a data-driven solution to automate the paper review process.
2. This paper introduces several methods to make its framework efficient and practical such as pairwise learning, pointwise inference; Review Tendency Signal etc.
3. The performance and generalizability of the framework appears to be good.

**Weaknesses:**

1. The relationship between NAIDv2 and NAIPv2 is confusing due to the naming. It appears that NAIDv2 is just the dataset used for learning the NAIPv2 framework.
2. The paper introduces models such as Review Tendency Signal, but it is unclear how good this probabilistic model is, compared to other alternatives, e.g., the review score might follow a Bradley–Terry model.
3. The experiment results on the neurips dataset is better than the previous method, but it is still unclear how it is useful for practical applications, e.g., can it guide the AC decision in assessing the paper quality in addition to human reviewers.

**Questions:**

What are some good application scenarios for NAIPv2?

---

> ### Author Response · Authors · 2025-11-19
> **Authors' Response to Reviewer hKKi**
>
> Thank you for your thoughtful assessment of our work and for recognizing its strengths in **efficient** and **practical generalizability**. We have carefully considered your concerns, and our point-by-point responses are as follows:
>
>
> ## Potential Naming Confusion
> **NAIDv2 indeed refers to the dataset.** The naming convention follows that the **"P"** in *NAIP* stands for **P**rediction, while the **"D"** in *NAID* stands for **D**ataset. We will revise the relevant descriptions to reduce potential ambiguity.
>
> ## Clarification of RTS and Bradley–Terry Formulation Validity
> In **Table 4**, we compared the impact of different proxies on overall model performance. Following the qFdP reviewer’s suggestion, we conducted new experiments using **citation count** as a proxy. Additionaly, in **Appendix D (Table 6)**, we further examined the effects of different loss constraints. These results collectively demonstrate the **robustness and effectiveness** of our proposed **RTS**  and the **Bradley–Terry formulation**.
>
> ## Clarification of Practical Applications
>
> As emphasized in the **General Response**, the core strength of our method is its ability to provide a **lightweight, efficient, and reasonably accurate ranking signal** in practical settings where PDFs are often unavailable and models rely solely on titles and abstracts. This makes our approach particularly valuable for **literature retrieval**, **literature recommendation**, **automatic survey generation**, and other forms of *automated scientific assistance*.
> Moreover, our method can serve as an **online reward signal** in reinforcement learning setups or provide **additional supervision** to enhance the quality-assessment capabilities of autoregressive reviewing models. Our overall design aims to balance **theoretical elegance** (as noted by reviewer g7oZ) with **practical effectiveness and scalability**.
>
> **Our work is not intended to replace or directly serve automatic reviewing.** As we clearly state in Appendix A, *"These scores are not intended, nor should they be used, to replace human peer review or editorial decision-making"*. We neither believe nor recommend that our model outputs be directly used for automated reviewing or as decision-making signals for ACs. ACs play an indispensable role in peer review, providing expert judgment, ethical oversight, and cross-paper coordination—capabilities that no automated system can currently replicate.
>
>
> We hope that our responses and the steps we have taken satisfactorily address your concerns. We fully respect your original evaluation, and we would appreciate a re-assessment should you find it appropriate.

---

### Author Response · Authors · 2025-11-12
**Acknowledgment of Reviews and Upcoming Clarifications**

We sincerely thank all reviewers for their thoughtful and detailed feedback on our submission.

We are carefully reviewing each comment and will provide a structured clarification soon, including additional analyses and theoretical explanations to address the raised questions.

We appreciate the constructive spirit of the reviews and look forward to the discussion.

---

### Author Response · Authors · 2025-11-19
**Authors' General Response - Part 2/2**

Below, we summarize the main concerns raised by reviewers, along with our responses and the actions we will take. More detailed replies will be provided separately to each reviewer.

| **Concern** | **Response** | **Action** |
|-|-|-|
| **(hKKi) Clarification of RTS** | RTS serves as a proxy signal rather than a standalone probabilistic model. As shown in Table 4, RTS outperforms four alternative proxy formulations. In Appendix D (Table 6), we further compare Bradley–Terry against other probabilistic formulations such as Thurstone–Probit and find that BT yields the best performance. | We will revise the relevant sections to make the design rationale clearer. |
| **(g7oZ) Suggestion about theoretical consistency** | Following the reviewer's recommendation, we conduct the suggested empirical experiments. Although the Thurstone–Mosteller model is theoretically more unified and elegant, its empirical performance, similar to the Thurstone probit attempt mentioned in the Appendix, was slightly weaker than the current Bradley–Terry based NAIPv2 implementation. Given that NAIPv2 often functions as the system’s foundational information filtering module, for which stronger performance is especially valuable, we prefer to keep the current approach unchanged. | We will add a discussion of the suggestion in the Appendix and include the additional experimental results in Tab. 6. |
| **(f3tL) Generalization to other LLMs** | To examine whether the strategy also works for other models, we conducted experiments on two additional LLM backbones. Both showed clear and consistent performance gains, confirming that the strategy is effective across different backbones. | We will add the additional results in the Appendix. |
| **(64Pf) Potentially limited applicability** | Our method is not designed for automated peer review; rather, its primary use is to support broader automated research workflows such as literature retrieval, recommendation, survey generation, and research assistants. The underlying idea can also benefit reinforcement learning and, potentially, auto peer-review systems themselves. Compared with PDF-based peer-review approaches, NAIPv2 offers a more lightweight and flexible solution suited to a wider range of applications. | We will add a dedicated subsection in the Appendix to discuss intended use cases. |
| **(64Pf) Potentially unfair comparisons** | As detailed in Appendix K, our evaluation protocol is fair: following common practice, we adopt a random split whose distribution closely matches that of DeepReviewer-13K, ensuring reasonable comparability. A strict apple-to-apple comparison is generally difficult because existing methods overlap in their training and test data due to the shared ICLR review source. Nevertheless, following the reviewer's suggestion, we additionally retrained NAIPv2 on the official DeepReviewer-13K training set; under identical data and settings, NAIPv2 performs on par with, and in some cases slightly better than, DeepReviewer. | We will incorporate this apple-to-apple experiment into Appendix K. |
| **(qFdP) More Proxy Indicators** | We did not use citation count because it varies across fields and years and is known to correlate weakly with reviewer scores [6]. Following the reviewer’s suggestion, we nevertheless tested citation-based proxies; their correlations were only similar to simple baselines (e.g., mean or mode), further supporting RTS as a stronger proxy signal.  | We will include the experimental results using citation count as a proxy. |

### References:
[1] Lu, Chris, et al. "The ai scientist: Towards fully automated open-ended scientific discovery." arXiv preprint arXiv:2408.06292 (2024).

[2] Wang, Yidong, et al. "Autosurvey: Large language models can automatically write surveys." NeurIPS (2024): 115119-115145.

[3] Paper Digest. arXiv Daily – Paper Digest. https://www.paperdigest.org/arxiv/

[4] Hugging Face. Daily Papers. https://huggingface.co/papers

[5] alphaXiv. Discover, Discuss, and Read arXiv Papers. https://www.alphaxiv.org/

[6] Cortes, Corinna, and Neil D. Lawrence. "Inconsistency in conference peer review: Revisiting the 2014 neurips experiment." arXiv preprint arXiv:2109.09774 (2021).

---

### Author Response · Authors · 2025-11-19
**Authors' General Response - Part 1/2**

We thank all reviewers for their time and constructive feedback. We appreciate the recognition of our work’s **clear motivation**, **principled confidence modeling**, **strong empirical performance**, and **the value of the curated large-scale dataset**. Before addressing main conerns, we would like to clarify the intended application scenarios of NAIPv2.

**NAIPv2 is NOT designed for automated peer review.**. As emphasized in both the first paragraph of the Introduction and Appendix N, the focus of NAIPv2 is to rapidly and cost-effectively predict a *quality signal* that can be used to rank a set of candidate papers for emerging tasks in the broader field of AI for Science, where automated peer-review methods are typically not suitable:

1. **Providing literature intelligence for cutting-edge AI for Science systems**

   *These systems (e.g., autonomous research systems like AI Scientist [1], or automated survey-generation systems like AutoSurvey [2]) typically rely on extensive literature retrieval. NAIPv2 can rapidly re-rank the retrieved papers within the same domain based on titles and abstracts, offering a lightweight quality-control mechanism that enhances overall system performance.*

2. **Serving as a cold-start foundation for various recommendation systems**

   *In the past few years, many paper recommendation services [3–5] have emerged, yet they all face a core challenge: ranking hundreds of newly released papers each day without citations, venue information, or reliable author features. NAIPv2 offers one of the few practical solutions by directly estimating paper quality from minimal, universally available inputs.*

3. **Supporting the Broader Research Ecosystem**

   *NAIPv2 is easy to use, and its pointwise inference speed reaches nearly 100 papers per minute even on a more common but less powerful GPU such as the RTX 3090. This enables NAIPv2 to provide a fast, even online, reward signal for reinforcement learning, thereby supporting self-evolution capabilities in models.*

Beyond the primary intended use cases, the method may also naturally extend to several downstream scenarios, with potential applications such as:

   - NAIPv2 could **enhance auto peer-review systems** by serving as a lightweight regression module that injects de-biased quality signals into autoregressive models without increasing significant training cost or affecting their language-modeling capabilities.
   - **Improving reviewer–submission matching systems** by rapidly ranking the quality of submissions within the same research area, thereby enabling automatic quality-stratified assignments that more effectively align reviewer expertise with submission *estimated* quality.
   - Serving as a **bibliometric statistical** analysis tool.
   - **Assisting authors** in refining their titles and abstracts. (as noted by reviewer qFdP).
   - ...

In most of above mentioned scenarios, several common constraints can be observed. For example, PDF files cannot always be reliably or legally obtained, and the title and abstract are often the only accessible information. Cross-domain comparisons are rarely needed, and the user’s waiting time is typically limited. With these practical considerations in mind, we integrated them into our theoretical formulation, algorithmic design, and experimental setup, ultimately presenting the NAIPv2 framework, which is shown to be effective and extensible.

---

### Meta-Review · Area_Chair_VYvi · 2026-01-10

**Summary:**

The reviewers initially had some significant divergence (2, 4, 8, 2, and 8). After the discussion, one 4 raised to 6, while others stayed the same (or could be due to the incidence that prevent anyone to engage in further discussions).

There are some very insightful comments raised by reviewers, and the authors have addressed most of them.

The authors put some sincere efforts in the rebuttal process, and provided additional experimentation. This has significantly improved the submission.

Therefore, I’d recommend the accept (poster) for this submission.

**Reviewer Concerns:**

Reviewers raised many concerns, such as

•	Comparison with citation-based proxy metrics
•	Improvement of presentation
•	Lack of reasoning capability
•	Lack of comparison with others (LLM finetuned, OpenReviewer), and not necessary better than existing work such as DeepReview
•	Dataset contribution is weak as has done by others
•	The discrepancy between the actual inputs (title and abstracts only) and the expectation that the overall insights should come from the whole paper
•	And many more…

But the authors have addressed most of them. There are still some open issues not fully addressed, such as so-called fair comparison with others work under the “same” setting. But this is much hard to do in practice, which I agree with the authors.

**Reviewer Scores:**

I hope the two reviewers (with 2 rating) might have changed their decision after seeing the discussion with other reviewers.

---

### Decision · Program_Chairs · 2026-01-26

Accept (Poster)